

Monthly element/Ca trends and inter chamber variability in two planktic Foraminifera
species: *Globigerinoides ruber albus* and *Turborotalita clarkei* from a hypersaline
oligotrophic sea
Noy Levy[1,2], Adi Torfstein[1,3], Ralf Schiebel[2], Natalie Chernihovsky[1,3], Klaus Peter
Jochum[2] ⸙, Ulrike Weis[2], Brigitte Stoll[2], Gerald H. Haug[2,4]
1) The Fredy & Nadine Herrmann Institute of Earth Sciences, Hebrew University of
Jerusalem, Jerusalem 91904, Israel.
2) Max Planck Institute for Chemistry, Hahn-Meitner-Weg 1, 55128 Mainz, Germany.
3) Interuniversity Institute for Marine Sciences, Eilat 88103, Israel.
4) Department of Earth Sciences, ETH Zurich, Sonneggstrasse 5, 8092 Zurich, Switzerland
⸙ Deceased, November 9, 2024
*Correspondence to*: Noy Levy (noy.levy2@mail.huji.ac.il)
Abstract
Environmental and biological factors influence the trace element composition (element/Ca) of
planktic foraminifer shells. Consequently, the element/Ca measured in these shells (tests) are
utilized as proxies to reconstruct past oceanic and climatic conditions. As single shell analyses
are increasingly used in paleoceanographic research it is important to understand how proxy
systematics change between species, individuals of the same species in a given population, and
among chambers of a single individual during its life cycle. Here we present a time series of
the chemical composition of planktic foraminifers retrieved using sediment traps between June
2014 and June 2015 at the northern part of the Gulf of Aqaba (aka Gulf of Eilat). Laser ablation
ICP-MS element/Ca measurements were performed on single shells and chambers of
*Globigerinoides ruber albus* and *Turborotalita clarkei*, collected monthly from five water
depths (120 m, 220 m, 350 m, 450 m, and 570 m). Sediment trap samples were paired with
corresponding data on water column hydrography and chemistry. Pooled means of measured
element/Ca display species-specific and element-specific behavior, with generally higher





values for *T. clarkei* phenotypes ('big' and 'encrusted') in comparison to *G. ruber albus*. Some
element/Ca values measured in water column specimens, such as Al/Ca, vary significantly
from core-top specimens. A unique finding is a prominent increase in element/Ca around
March-April 2015, during maximum water column mixing, mostly apparent in *T. clarkei* and
to a lesser extent in *G. ruber albus*. This spring element/Ca increase is observed in most
measured elements and is further associated with an increase in inter-chamber variability
(ICV). Inter-chamber element/Ca patterns show element enrichment/depletion in the most
recently precipitated (final, F0) chamber in comparison to the older chambers (penultimate (F-
1), antepenultimate (F-2), etc.). Element/Ca in F0 may also be less sensitive to surrounding
environmental conditions. For example, the Mg/Ca of the F-1 and F-2 chambers of *G. ruber*
*albus* display a positive relationship with mixed layer temperatures while F0 does not. To
overcome this effect, we suggest using pooled means from non-F0 fractions as environmental
records and paleo proxies.
These results highlight the complexity of proxy systematics that rises from the variability in
element/Ca measured among different species and between chambers, caused by ecological
conditions and other processes in the water column including physical, chemical, and
biological effects.

1. Introduction
1.1 Planktic foraminifera as traces of the past environment
Planktic Foraminifera (PF) shells are useful archives for studying the history of Earth's
climate and oceans, as their calcareous shells reflect the environmental conditions during their
formation (Berggren et al., 1995; Rosenthal, 2007; Schiebel & Hemleben, 2017; Kucera, 2007;
Katz et al., 2010; Gupta, 1999; Davis et al., 2020, and others). Various element/Ca measured
in PF tests have been closely linked to ambient seawater temperature (e.g., Mg/Ca; Rosenthal
et al., 2004), salinity (e.g., Na/Ca; Mezger et al., 2016; Gray et al., 2023), $p$H and the carbonate
system (e.g., B/Ca; Babila et al., 2014; Henehan et al., 2015; Haynes et al., 2019), productivity
(e.g., Ba/Ca; Fritz-Enders et al., 2022), and chemical weathering (e.g., Ti/Ca; Amaglio et al.,
2025, among others. In the past, the use of these proxies relied on bulk analysis of the entire
shell or multiple shells. However, in recent years there has been an increase in the use of high-
resolution analytical techniques, such as Laser Ablation (LA) ICP-MS and electron microprobe
analyses in paleoceanographic studies (Davis et al., 2020). The element/Ca measurements of



single specimens (Individual Foraminifer Analysis, IFA) revealed high variability between
individuals of the same population as well as significant intra-shell variability (i.e., inter
chamber variability, ICV) (Sadekov et al., 2008; Fehrenbacher et al., 2020; Hupp &
Fehrenbacher, 2024; Fischer et al., 2024; Davis et al., 2020, and references therein). The
associated changes in the geochemical signatures of PF shells are poorly understood and
despite the analytical advancements, there are still knowledge gaps in our understanding of
proxy systematics in single shell and single chamber of PF species although they are potentially
related to the observed shell and chamber element/Ca variability, the life cycles and
reproductive modes of many species, as they calcify their shell chambers one at a time. There
is also a lack of detailed description and understanding of proxy systematics in description of
the dynamics of small-sized species such as *T. clarkei*, which have been largely overlooked in
previous studies despite their significant contribution to the settling PF tests (export flux), as
observed in the northern Red Sea (Chernihovsky et al., 2018). Furthermore, specific marine
regions, such as in oligotrophic, subtropical basins, particularly in deep-water column
environments, are not well-established in terms of their spatial and temporal dynamics
(Schiebel & Hemleben, 2017).
1.2 Planktic Foraminifer population dynamics in The Gulf of Aqaba
The Gulf of Aqaba (GOA) is considered an open ocean proxy environment (Chase et al.,
2011). It is an oligotrophic basin where the main lithogenic flux is derived from dust. During
summer (April-September), a ~200 m deep thermocline separates nutrient-depleted surface
waters (~25°C) from the nutrient-rich deep layer (~21°C). In winter/spring (October-April),
the thermocline gradually erodes due to surface cooling (Figs. 1a and 1e; Meeder et al., 2012),
which can lead to the development of a deep mixed layer. Although the depth of the mixed
layer varies annually with climatic conditions, the long-term mean mixing depth is
approximately 300-400 m, and deep mixing can extend to the sea floor while it typically
reaches maximum depth by late March. The regional terrestrial climate is hyper-arid (mean
annual rainfall <30 mm) and the main sources for terrigenous material to the GOA are dust
storms originating from the Sahara and Arabian Deserts, as well as rare localized floods (Katz
et al., 2015; Chase et al., 2011; Ganor et al., 2001; Torfstein et al., 2017).





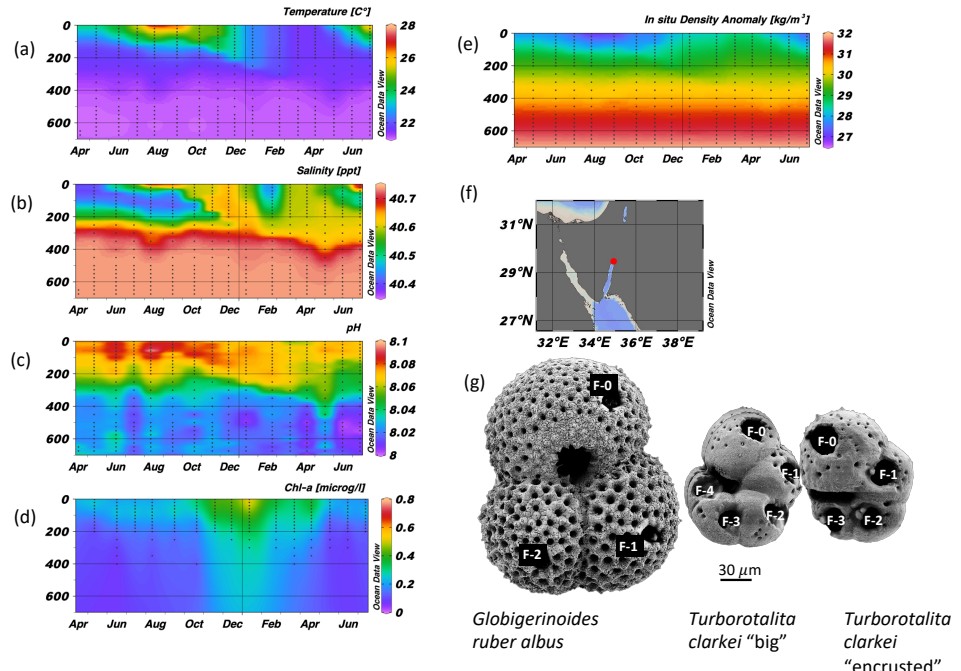

Figure 1: Time series of temperature (a), salinity (b), *p*H (c), Chlorophyll-a concentration (d), and in situ density anomaly (e), measured in the Gulf of Aqaba between April 2014 and June 2015 by the National Monitoring Program (NMP, Shaked & Genin, 2016). Y-axis is depth (m); A map of the Gulf of Aqaba (f); and (g) scanning electron micrographs of the three morpho-species (exhibiting ablation holes in each chamber (labelled), from Levy et al., 2023).

Planktic foraminifera fluxes in the GOA demonstrate strong seasonality, with low fluxes during the summer months, gradually increasing during the autumn-winter, coeval with decreasing sea-surface temperatures and deepening of the mixed layer in the GOA that drives advection of nutrient-replete subsurface waters into the mixed layer. This in turn triggers an increase in primary productivity, expressed by enhanced chlorophyll-a concentrations and high PF fluxes (Chernihovsky et al., 2018, 2020).

Spinose species constitute the majority of the PF assemblage. The smaller size fraction, 63-125 μm, is 86% from the total flux and is dominated by *T. clarkei*. The 125-500 μm size-fraction (~13 %) is dominated by the species *G. ruber albus*, while less than 1% of the shells are in the range of 500-1000 μm, dominated by *O. universa* (Chernihovsky et al., 2018).

*Globigerinoides ruber albus* and *T. clarkei* inhabit different dwelling-depths and have diverse life strategies. *Globigerinoides ruber albus* is a surface dweller and is photo-symbiont



bearing, while *T. clarkei* tends to dwell below the mixed layer depth and is barren of photo-
symbionts (Rebotim et al., 2017; Schiebel & Hemleben, 2017; Levy et al., 2023). Furthermore,
it has been suggested that *G. ruber albus* and *T. clarkei* do not share the same dietary
preferences: *G. ruber albus* being more carnivorous than the detritivorous *T. clarkei* which may
forage at the exported matter below the pycnocline (Schiebel & Hemleben, 2017). In the GOA,
*T. clarkei* has two phenotypes: *T. clarkei* 'big', which all of its test chambers are fully
recognizable and their surface is relatively smooth and *T. clarkei* 'encrusted' with a less smooth
shell surface and is smaller than the 'big' type (Levy et al., 2023).

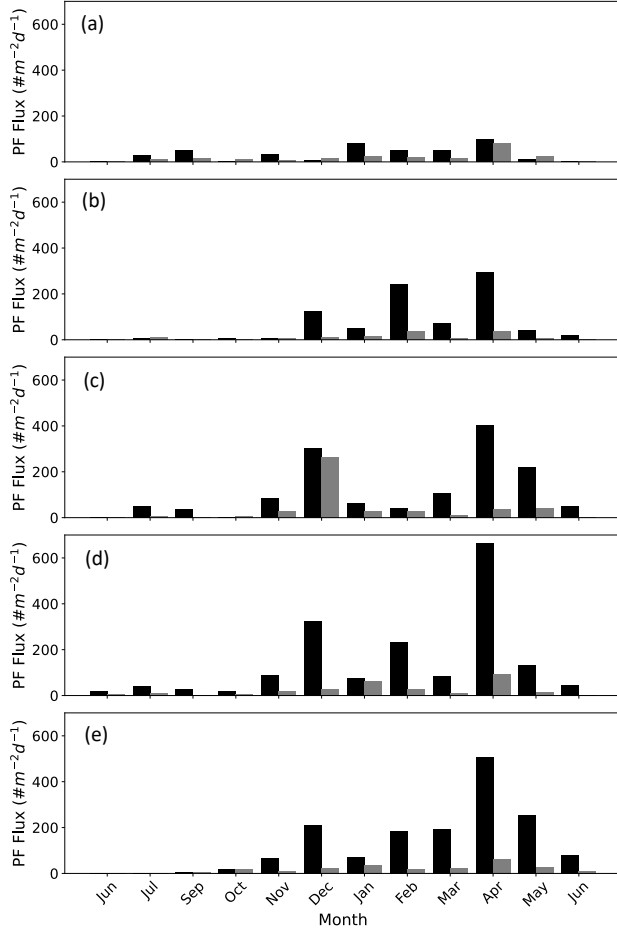


Figure 2: PF fluxes in the Gulf of Aqaba between June 2014 and June 2015 presented according
to the size fractions >63-125 µm (black bars) and >125-500 µm (grey bars) at the different
sediment trap depths a) 120 m, b) 220 m, c) 340 m, d) 450 m, and e) 570 m. Data from
Chernihovsky et al. (2018).



127  In this study, we investigate the range of element values, year-round trends and inter-
128 chamber element/Ca variability in *G. ruber albus* and *T. clarkei* tests collected in sediment
129 traps at various water column depths from the GOA. We assess whether the chambers record
130 temporal-seasonal patterns, and the implications for using single chamber data for geochemical
131 proxies (Mg/Ca, B/Ca, Na/Ca). Understanding inter-chamber variability sheds light on
132 biomineralization processes and environmental factors that occur during different stages of the
133 organism's life cycle. This in turn improves the calibration of element/Ca as proxies for reliable
134 reconstruction of past oceanic and climatic conditions. Focusing on PF from the GOA provides
135 critical insights into the use of foraminiferal element/Ca as proxies in a warm and hyper-saline
136 oligotrophic environment.

137

138 2. Methodology

139

140 2.1. Sampling and oceanographic data

141  A bottom-tethered mooring has been deployed continuously since January 2014 near
142 Station A, northern GOA (29° 28'95' N, 34° 56'22' E, ~605 m water depth) (Torfstein et al.,
143 2020). Five KC Denmark cylinder sediment traps were mounted vertically and located at
144 depths of 120 m, 220 m, 350 m, 450 m, and 570 m. The trap samples were collected at a
145 monthly resolution. Furthermore, PF samples from the sediment interface were collected using
146 a box core ('core top'). Further detailed description of the mooring, sampling, sample
147 processing, and trapping efficiencies can be found in Chernihovsky et al. (2018) and Torfstein
148 et al. (2020). Here, we report the findings derived from the PF tests collected between June
149 2014 and June 2015. Water column physical and chemical parameters are routinely collected
150 at Station A by the Israel National Monitoring Program (NMP, Shaked & Genin, 2016). This
151 includes sea surface and water column temperature (°C), salinity, oxygen concentration
152 (μmol/l), alkalinity (meq/kg), $p$H, and chlorophyll-a concentration (μg/l). The mixed layer
153 depth (MLD) is defined as the water depth where the density anomaly (σ0) is equal to, or above,
154 the water density of the surface water column plus a density threshold of 0.125 kg/m$^3$ (Sprintall
155 & Tomczak, 1992).

156 2.2. Species classification and preparation for LA-ICP-MS






We examined the shell chemical properties of two flux dominating PF species *T. clarkei*
and *G. ruber albus* (i.e., sensu stricto, white). For *T. clarkei* we examined two morphotypes:
'big' and 'encrusted'. Identification and nomenclature of the PF taxa followed Schiebel &
Hemleben (2017), Morard et al. (2019), and Brummer & Kucera (2022). Three individuals
were picked from each sediment trap depth during each month between June 2014 and June
2015. Preliminary preparation and cleaning steps are detailed by Chernihovsky et al. (2018).
Reductive and oxidative cleaning had been avoided to retain original signals related to the
different encrustation processes and preserve all calcite layers added to the shell during
ontogeny (Schiebel & Hemleben, 2017; Jochum et al., 2019). Specifically, the shell *T. clarkei*
is prone to loss of material during reductive and oxidative treatment as it has very thin shells
with a width ranging 1.9-3.6 μm (Levy et al., 2023). Single chamber measurements were
performed to asses inter chamber variability (ICV), on individual shells (individual foraminifer
analysis; IFA) using Laser Ablation Inductively Coupled Plasma Mass Spectrometry (LA-ICP-
MS) on 156 specimens in total. Samples were glued to glass slides using a methyl-hydroxy-
propyl-cellulose (MHPC 1:100), positioned with the umbilical side up.

2.3. LA-ICP-MS and data processing
Analyses of the calcium-normalized elements for B, Na, Mg, Al, Ti, Mn, Fe, Co, Sr, Ba,
Nd, Pb, Th, and U were conducted using a 200 nm wavelength NWR femtosecond (fs) LASER
system from ESI, combined with a sector-field Thermo Element-2 ICP mass spectrometer
(Jochum et al., 2014). Measurements were performed using a 15 Hz pulse repetition rate (PRR),
at low fluence ($0.1$–$0.6$ J/cm$^2$), and 18 seconds dwelling time. A 30 μm diameter spot size was
selected, as it is the maximum diameter for analysis fitting in a single chamber of the small *T.*
*clarkei*. The microanalytical synthetic reference material MACS-3 for carbonate, NIST-612,
and NIST-610 were used for calibration. NIST-612 was used for the tuning of the ICP-MS
(Jochum et al., 2019).
The measurement precision (1 relative standard deviation in percent; 1 RSD) yield
uncertainties for references materials between ~ 5-17 % for the calcium-normalized elements
(Supplementary table 1). Single spot measurements were made on each chamber of the
individual shells. Chambers are labelled F0 (final chamber), F-1 (final minus one), F-2, and so
on, for the penultimate, antepenultimate, and further chambers, respectively. We calculated
averages and standard deviations of element/Ca of single individuals (calculated from all single



chamber element /Ca in one shell) and relative standard errors of element/Ca of pooled
measurements for a specific morphotype.
3. Results:

3.1. Depth-averaged values of element/Ca measured in *G. ruber albus* and *T. clarkei*
shells using LA-ICP-MS

Generally, the means of Mg/Ca, Sr/Ca, B/Ca, Na/Ca and Ba/Ca in *G. ruber albus* indicate that
the composition of tests, from most water depths is similar to that of core-top samples (Figs.
3a-3d, 3j). In contrast, Al/Ca, Ti/Ca, Mn/Ca, Fe/Ca, Nd/Ca, Th/Ca, and U/Ca (Figs. 3e-3i, 3k,
3m, 3n) in the tests from sediment interface were higher than in the water column, and lower
in case of Co/Ca and Pb/Ca (Figs. 3i, 3l).



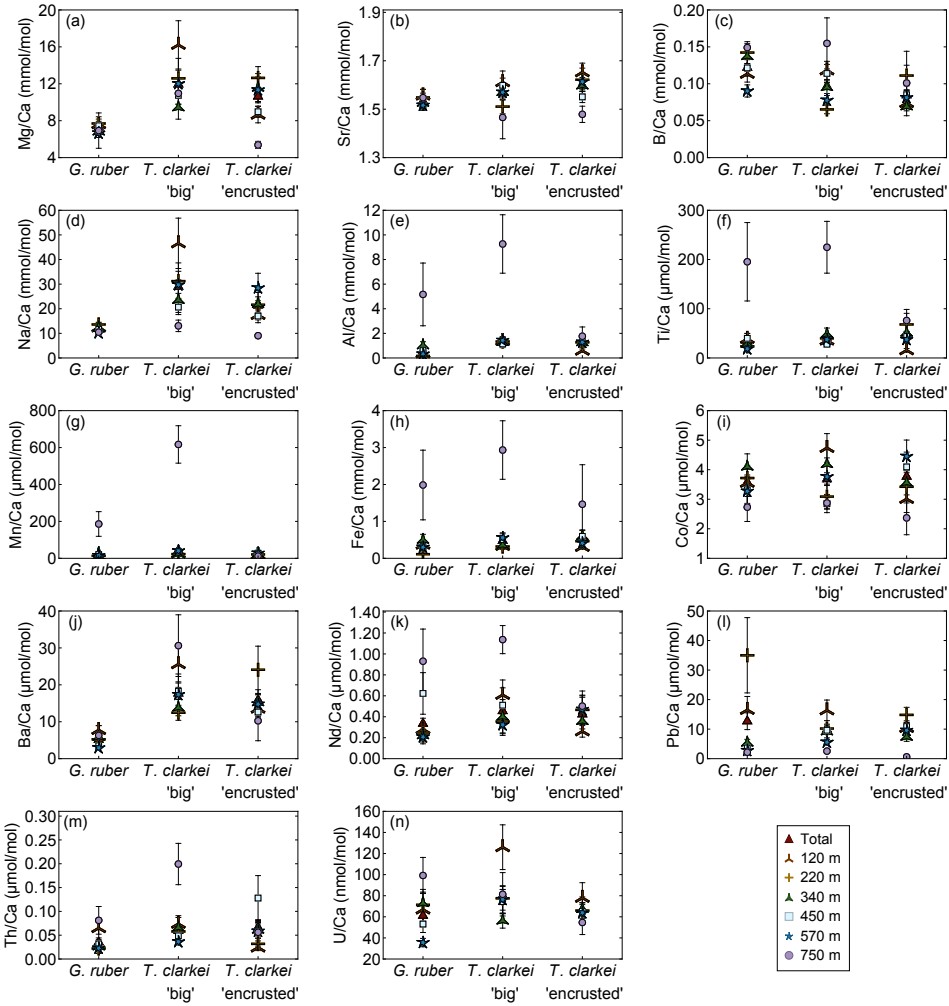

Figure 3: Pooled mean values of the calcium-normalized element ratios of *G. ruber albus*, *T. clarkei* 'big' and *T. clarkei* 'encrusted' shells, derived from sediment traps located at different water depths (120 m to 570 m) and a core top sample (750 m) from the Gulf of Aqaba. Error bars represent 1 sigma relative standard error ($SD/\sqrt{n}$).

Furthermore, *T. clarkei* tends to demonstrate higher values and higher variability compared to *G. ruber albus*. Compared to the core-top samples, *T. clarkei* from the water column also exhibit relative enrichment in Al/Ca, Ti/Ca, Mn/Ca, Fe/Ca, Nd/Ca, B/Ca, and Th/Ca (*T. clarkei* 'big'), and depletion in Co/Ca, Pb/Ca, Sr/Ca, and Mg/Ca (*T. clarkei* 'encrusted') (Fig. 3).

3.2. Shell-bound element/Ca time series trends in *G. ruber albus* and *T. clarkei* shells




Pooled mean values of Mg/Ca in *G. ruber albus* taken from all water column depths in the
GOA reflect MLD temperatures (Fig. 10). Single chamber Mg/Ca over water column depths
in *G. ruber albus* range between 2.01 mmol/mol (340 m; June 2015) and 18.49 mmol/mol (340
m; July 2014), with lower/higher values during winter/summer months, respectively (Figs. 4a-
4e). A unique observation is an increase in Mg/Ca seen during spring (March-April), i.e.,
months with maximum surface water column mixing, at some water depths (220 m, 340 m,
450 m; Figs. 4b-4d). Accompanied with the Mg/Ca increase is a clear increase in ICV as
evident by the divergence of chamber values. Generally, it appears that Mg/Ca is lower in F0
chambers (orange dotted line) compared to preceding chambers. Mg/Ca in *T. clarkei* 'big'
range between 4.00 mmol/mol (340 m; June 2015) and 77.02 mmol/mol (220 m; March 2015)
and between 4.06 mmol/mol (570 m; December 2014) and 51.22 mmol/mol (120 m; April
2015) in *T. clarkei* 'encrusted', respectively.

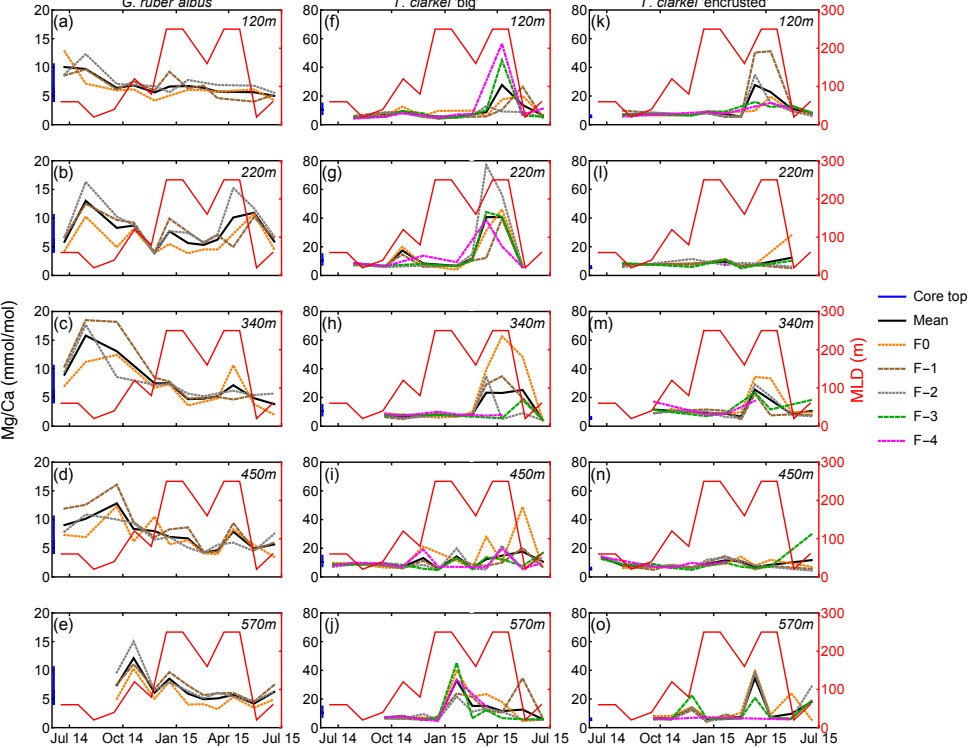

Figure 4: Time series of Mg/Ca values measured from the shells of *G. ruber albus* (a-e), *T.*
*clarkei* 'big' (f-j) and *T. clarkei* 'encrusted' (k-o), derived from sediment traps located at





different water depths (120-570 m). Mg/Ca of core top are marked by a blue bar along the left
y-axes.

Sr/Ca in *G. ruber albus* range between 1.25 mmol/mol (570 m; January 2015) and 2.27
mmol/mol (340 m; November 2014) (Figs. 5a-5e). Single chamber Sr/Ca in *T. clarkei* 'big'
range between 0.94 mmol/mol (340 m; January 2015) and 2.76 mmol/mol (220 m; April 2015)
and for *T. clarkei* 'encrusted' between 0.54 mmol/mol (340 m; April 2015) and 2.92 mmol/mol
(570 m; June 2015), respectively (Figs. 5f-5j, and 5k-5o). *Turborotalita clarkei* 'big' and *T.*
*clarkei* 'encrusted' display more ICV than *G. ruber albus*, with peaking Sr/Ca in numerous
chambers around April 2015 (Figs. 5f-5o). During the spring months of 2015, Sr/Ca values
range between 1.45-2.04 mmol/mol in *G. ruber albus*, 1.32-2.76 mmol/mol in *T. clarkei* 'big'
and 0.54-2.27 mmol/mol in *T. clarkei* 'encrusted', respectively (Fig. 5; Fig. S1).

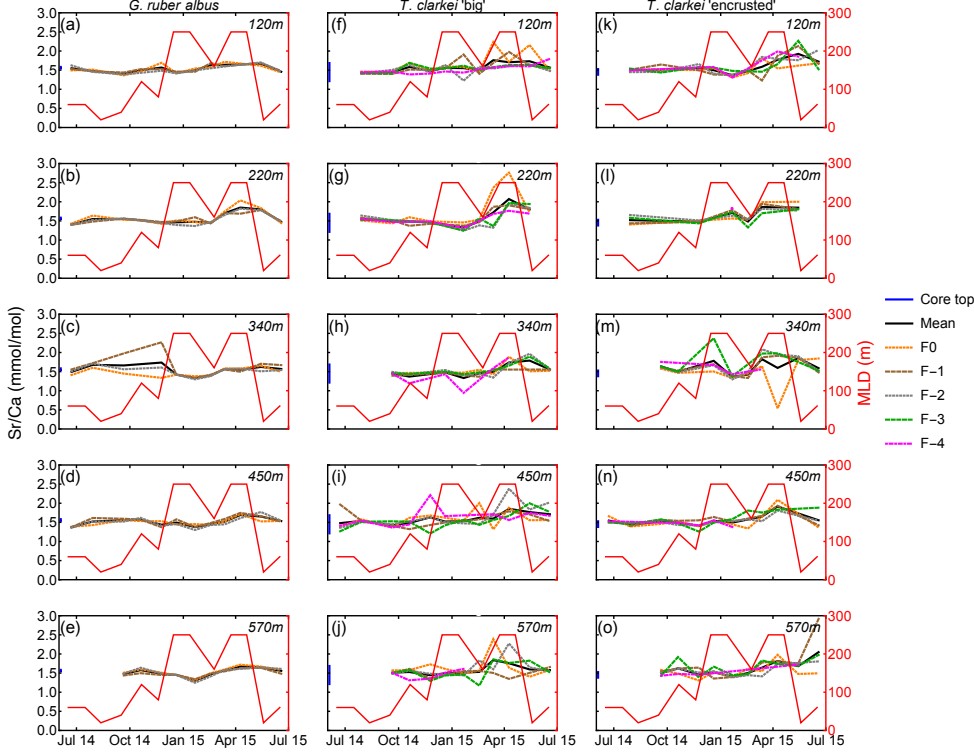

Figure 5: Time series of Sr/Ca values measured from the shells of *G. ruber albus* (a-e), *T.*
*clarkei* 'big' (f-j) and *T. clarkei* 'encrusted' (k-o), derived from sediment traps located at
different water depths (120 m – 570 m).






B/Ca values range between 0.03 mmol/mol (570 m; January 2015) to 0.35 mmol/mol (120 m;
June 2015) in *G. ruber albus*, with higher values during summer and spring and lower values
during the winter (Figs. 6a to 6e). B/Ca measured in the final chamber, F0, are systematically
lower compared to F-1 and F-2 values. Unlike most other element ratios, B/Ca values in both
phenotypes of *T. clarkei* are similar to the range measured in *G. ruber albus*. In both *T. clarkei*
phenotypes, lower B/Ca values were measured during the winter months, most prominently in
January. The B/Ca values of *T. clarkei* 'big' range between 0.01 mmol/mol to 0.53 mmol/mol
with some higher values during spring (Figs. 6f to 6j). B/Ca values in *T. clarkei* 'encrusted'
range between 0.01 mmol/mol to 0.47 mmol/mol. Generally, B/Ca ICV is higher in *T. clarkei*
than *G. ruber albus*, especially during spring.

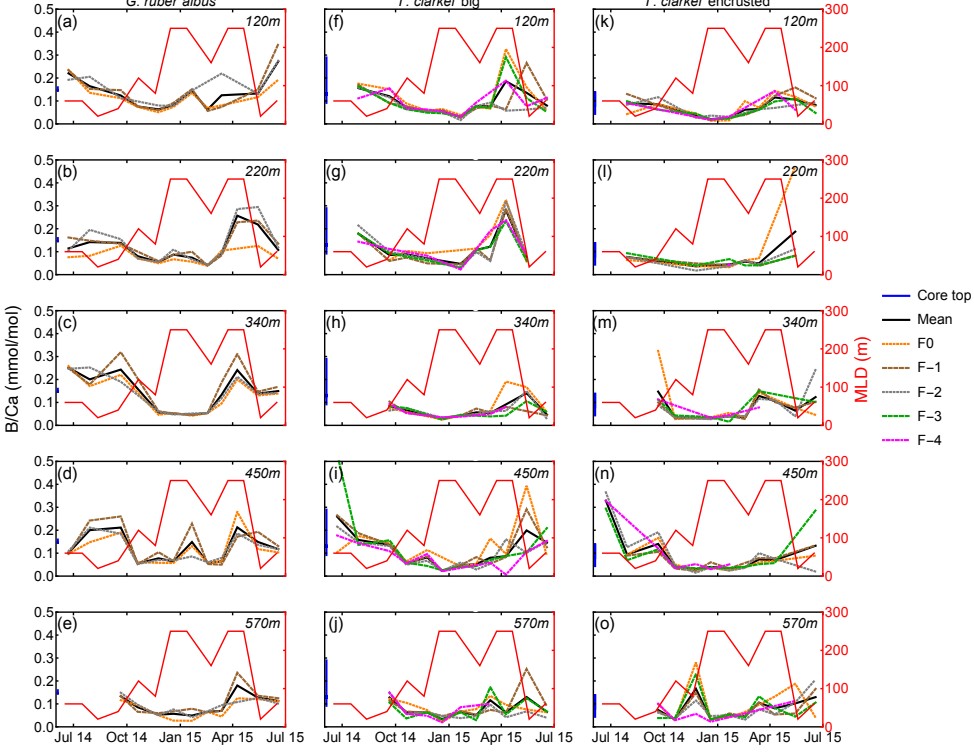


Figure 6: Time series of B/Ca values measured from the shells of *G. ruber albus* (a-e), *T. clarkei*
'big' (f-j) and *T. clarkei* 'encrusted' (k-o), derived from sediment traps located at different traps
from120 m to 570 m water depths.




Na/Ca in *G. ruber albus* ranges between 6.60 mmol/mol (220 m; June 2014) to 64.14
mmol/mol (220 m; April 2015) with a median value of 10.43 mmol/mol (Fig. 7; Fig. S1). Na/Ca
in *T. clarkei* 'big' ranges from 6.23 mmol/mol (570 m; September 2014) to 426.54 mmol/mol
(220 m; March 2015) with a median value of 12.33 mmol/mol. Na/Ca in *T. clarkei* 'encrusted'
ranges between 5.43 mmol/mol (570 m; September 2014) to 176.91 mmol/mol (570 m; March
2015) with a median value of 12.41 mmol/mol. *Globigerinoides ruber albus* has a low ICV
during spring, while *T. clarkei* 'big' and 'encrusted' phenotypes display higher ICV during the
same time interval. All morphotypes include significant excursions in Na/Ca with high values
in *G. ruber albus* during January and April at 220m (Fig. 7b), and high Na/Ca in both *T. clarkei*
phenotypes at multiple depths and seasons (Figs. 7f-7j and 7k-7o). In particular, *T. clarkei*
phenotypes show significant Na/Ca excursions during March-April and ICV (Figs. 7f-7o).

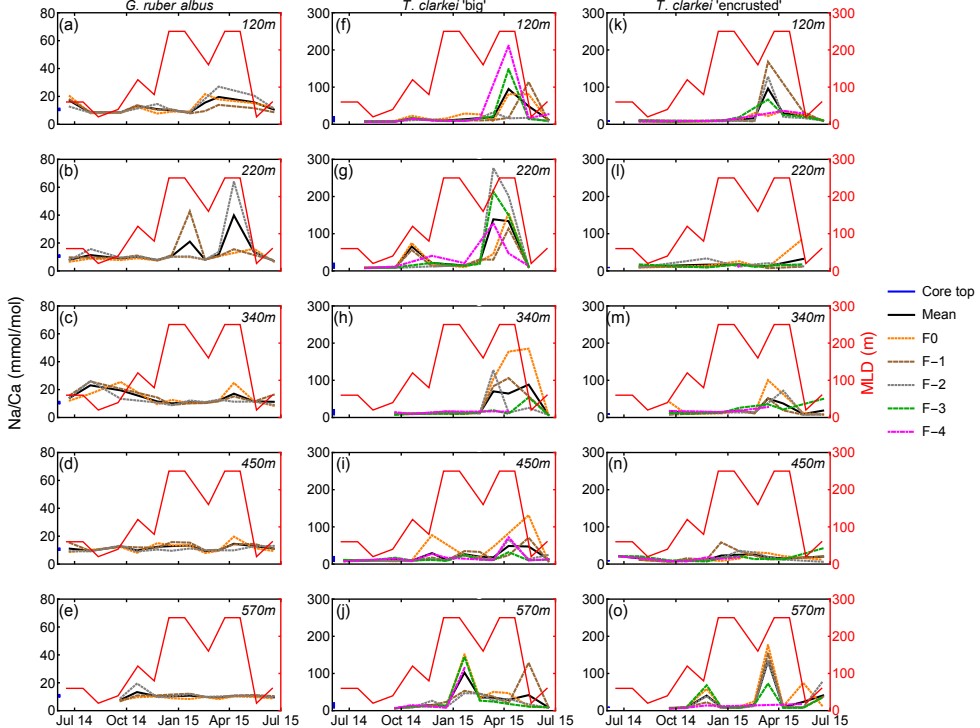

Figure 7: Time series of Na/Ca values measured from the shells of *G. ruber albus* (a-e), *T.*
*clarkei* 'big' (f-j) and *T. clarkei* 'encrusted' (k-o), derived from sediment traps located at
different water depths (120 m – 570 m).

Ba/Ca in *G. ruber albus* ranges from 0.73 μmol/mol (120 m; November 2014) to 36.81
μmol/mol (340 m; June 2015). Ba/Ca in *T. clarkei* 'big' ranges from 0.39 μmol/mol (120 m;



June 2015) to 246.54 µmol/mol (450 m; March 2015). Ba/Ca in *T. clarkei* 'encrusted' ranges
from 0 µmol/mol (April 2015) to 171.41 µmol/mol (340 m; March 2015) (Fig. 8; Fig. S1). The
three morphotypes display varied ICV, although *T. clarkei* shows more prominent ICV during
spring months (Figs. 8f-8o) than *G. ruber albus* (Figs. 8a-8e).

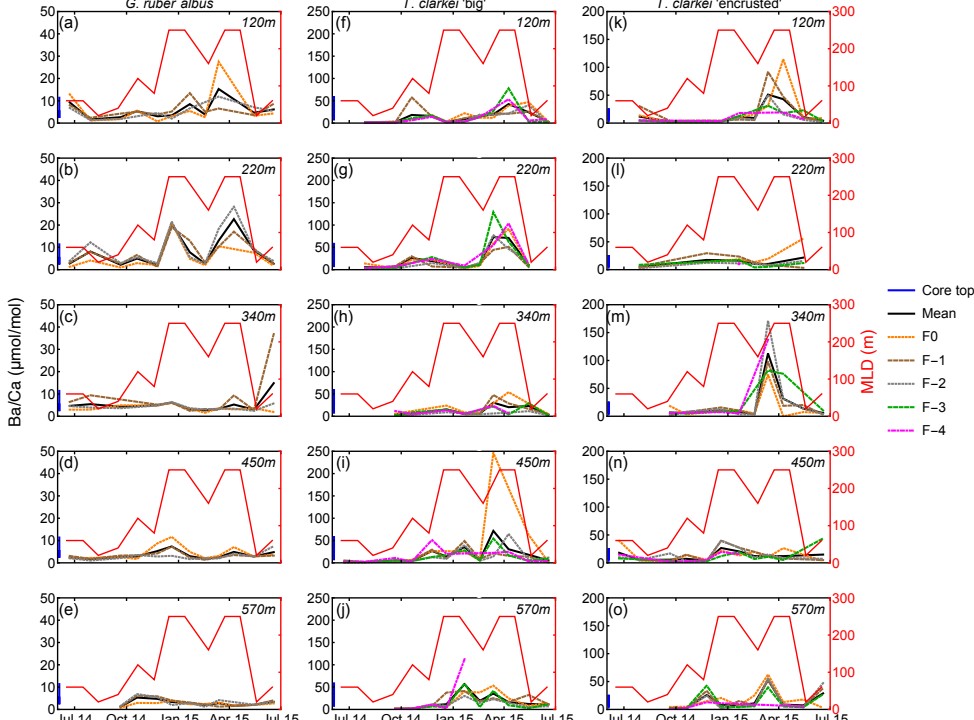

Figure 8: Time series of Ba/Ca values measured from the shells of *G. ruber albus* (a-e), *T.*
*clarkei* 'big' (f-j) and *T. clarkei* 'encrusted' (k-o), derived from sediment traps located at
different water depths (120 m – 570 m).

3.3. Relationships between element/Ca of the different PF species in the GOA

A Spearman correlation matrix was applied to assess the numerical relationships of the
element/Ca in the three analyzed PF phenotypes (Fig. 9; Tab. S2). The *T. clarkei* types exhibit
similar pattern of relationships, with minor differences mainly in correlation strength (Fig. 9a,
9b). In general, *T. clarkei* shows more significant relationships than *G. ruber albus*, while, *G.*
*ruber albus*, display different relationships to those of the two *T. clarkei* types. In *T. clarkei,*
Mg/Ca displays relatively strong relationships with Na/Ca, Ba/Ca, and Al/Ca (Fig. 9b, 9c).
Sr/Ca, B/Ca, Co/Ca and Nd/Ca do not display significant relationships to other elements in *G.*
*ruber albus* as well as in *T. clarkei* 'big' and *T. clarkei* 'encrusted'.





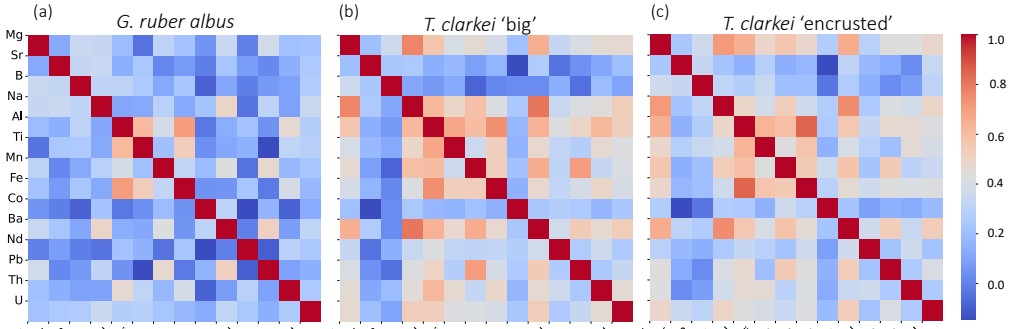


Figure 9: Spearman correlation Matrix of the different trace element Ca normalized abundances in *G. ruber albus* (a), *T. clarkei* "big" (b) and, *T. clarkei* "encrusted" (c).


For both *T. clarkei* 'big' and *T. clarkei* 'encrusted', Na/Ca significantly correlates with Al/Ca, Mn/Ca, and Ba/Ca, the later showing the strongest relationships in *T. clarkei* 'big' (r = 0.82, Fig. 9b; Tab. S2). Relationships between Al/Ca, Ti/Ca, Mn/Ca, Fe/Ca, Ba/Ca and Th/Ca are generally stronger in *T. clarkei 'big'* than in *T. clarkei 'encrusted'*, except for Al/Ca and Fe/Ca, which are stronger related in *T. clarkei 'encrusted'* (r = 0.85; Tab. S2) than in *T. clarkei 'big'* (r = 0.74; Tab. S2). Unlike *G. ruber albus*, the U/Ca in *T. clarkei* exhibit relatively strong relationships with Ba/Ca, Na/Ca and Al/Ca (in *T. clarkei* 'big') and Mg/Ca (in *T. clarkei* 'encrusted', Fig. 9c) while in *G. ruber albus*, U/Ca is poorly related to the other elements (Fig. 9a).

4.  Interpretation

4.1 Inter chamber variability (ICV)

Shell-bound element/Ca display varying trends across different chambers depending on the specific element ratios, and varying over water depth and time (Figs. 4-8). Typically, most PF reproduction-cycles span about a month with individual chambers forming within several hours (Bé et al., 1977), while the time interval between chamber formation can range from hours to weeks (Schiebel & Hemleben, 2017, and references therein). Setting aside the March-April time-interval where PF shells show exceptionally high ICV, *G. ruber albus* generally exhibits lower values (e.g., Mg/Ca, B/Ca), and less ICV compared to *T. clarkei* 'big' and 'encrusted'. The residence of *G. ruber albus* in the relatively homogenous and narrow living environment



in the surface mixed layer (Schiebel & Hemleben, 2017; Thirumalai et al., 2014; and others),
could explain relatively lower ICV. In contrast, *T. clarkei* dwell in the dynamic region
near/under the thermocline (Schiebel & Hemleben, 2017; Levy et al., 2023) over a wider
dwelling depth horizon, and may experience more heterogeneous environmental conditions
which may result in higher ICV.

The secondary crust observed on *T. clarkei* 'encrusted' morphotypes, which covers all
chambers of the tests analysed here, does not significantly alter element/Ca when compared to
*T. clarkei* 'big', unlike the crust of *Neogloboquadrina dutertrei* (Jonkers et al., 2012). This
suggests that the secondary calcite layer in *T. clarkei* 'encrusted' does not play a major role in
element incorporation or ICV and is affected by the same mechanisms which control the
formation of the ontogenetic calcite, and thus would facilitate application of our finding to the
interpretation of fossil *T. clarkei* 'encrusted' in paleoceanography and paleoclimate
reconstructions.

The ultimate chamber (F0) presents different systematics compared to the preceding
chambers in both *T. clarkei* and *G. ruber albus* (Fig. S11). In *T. clarkei* (both 'big' and
'encrusted'), the F0 typically exhibits higher values of B/Ca, Na/Ca, Mg/Ca, and Al/Ca
compared to the previous chambers. In contrast, *G. ruber albus* displays relatively lower values
in F0 for the same ratios highlighting species-specific differences in chamber formation (Fig.
S11). Interestingly, Sr/Ca does not follow the same pattern. In *T. clarkei* 'big' the Sr/Ca
distribution mirrors the trends of other elements, while F0 in *G. ruber albus* and *T. clarkei*
'encrusted' shows an even distribution of Sr/Ca, likely reflecting the relatively constant Sr/Ca
values in the water column during the lifespan of a single test. These observations in *G. ruber*
*albus* are consistent with previous studies that measured Mg/Ca in individual chambers (Bolton
et al., 2011; Davis et al., 2020; Fischer et al., 2024). The contrasting systematics of F0 leading
to elevated ICV in the ultimate chamber compared to the previous chambers was previously
suggested to be associated with a chamber wall that is not fully calcified (Schiebel &
Hemleben, 2017; Bolton et al., 2011; Fischer et al., 2024). Differences in F0 systematics
between *T. clarkei* and *G. ruber albus* could be driven by species-specific calcification
processes, though further research is needed to clarify this issue.  Additionally, it is important
to consider potential biases in small chambers such as F-4 in *T. clarkei* morpho-species, where
methodological challenges (e.g., laser spots hitting sutures) may skew element/Ca
measurements. Consequently, we conclude that the exclusion of F0 and F-4 will enhance the
reliability of reconstructions of the marine environment in studies of downcore records.



4.2 Relationships of element ratios of the three PF morphotypes

The contrasting results of the correlation matrixes of the three morphospecies, suggests species-
specific mechanisms while calcifying their shells. The Mg/Ca in *T. clarkei* which strongly
correlates with Na/Ca, Ba/Ca, and Al/Ca (Fig. 9c), suggests more than one environmental
process affects Mg/Ca in the tests as the other element/Ca are considered proxies to different
environmental characteristics such as salinity, productivity, and terrigenous input (Chang et al.,
2015; Mesa-Fernández et al., 2022; Beasley, et al., 2021). Similar to *G. ruber albus*, in the *T.*
*clarkei* types Sr/Ca, B/Ca, Co/Ca and Nd/Ca do not display statistically significant
relationships to other elements making them suitable proxies for distinct and independent
environmental properties.
In *G. ruber albus*, Mg/Ca, Sr/Ca and B/Ca show no significant relationships with other element
ratios, indicating that independent processes likely govern their proxy systematics (Fig. 9c).
Similarly, Co/Ca, Nd/Ca and U/Ca also do not correlate with other element/Ca. While Na/Ca
and Ba/Ca exhibit some degree of correlation, as do Mn/Ca and Pb/Ca, the lithophilic elements,
Al/Ca, Ti/Ca, which are considered proxies for terrigenous dust input (Chang et al., 2015;
Mesa-Fernández et al., 2022; Beasley, et al., 2021), as well as, Fe/Ca, and Th/Ca, all show a
relative strong correlation. Their correlation implies they can be used together for
reconstructing terrigenous input to the water column. Among the lithophilic elements, Th/Ca
display a relatively weaker relationship, suggesting a potential effect of additional processes
such as scavenging (Anderson et al., 1983; Francois et al., 2004; Costa et al., 2020).

4.3 Mg/Ca as a proxy for sea surface temperature

Shell-bound Mg/Ca of calcareous foraminifera have been extensively utilized as a paleo-
thermometer (e.g., Nürnberg et al., 1996; Sadekov et al., 2009). Many of these Mg-temperature
calibrations rely on whole-test or pooled-mean Mg/Ca values to reconstruct past sea surface
temperatures (Spero et al., 2003; Ganssen et al., 2010; and others). Several studies have
measured intra-test and inter-test Mg/Ca in an effort to produce Mg-temperature calibrations
using single chamber measurements of *G. ruber* (Sadekov et al., 2008; Bolton et al., 2011;
Davis et al., 2020; Levy et al., 2023; Fischer et al., 2024). Previous work on sediment trap-
derived specimens of *T. clarkei* and *G. ruber albus* from the GOA indicated that *T. clarkei* is
not suitable for temperature reconstructions, due to its presumed deep dwelling-depth below
the thermocline together with its high sensitivity to water column mixing events. However,
while *G. ruber albus* shows exceptionally high pooled mean Mg/Ca values in the GOA in
comparison to other ocean regions, it also exhibits seasonal variations that indicate effective



applicability as a paleothermometer (Levy et al., 2023). Due to the high seawater salinity of
the GOA, a local calibration curve was proposed (Eq. 1; Levy et al., 2023).

$$\frac{\text{Mg}}{\text{Ca}} = 0.39(\pm0.30) \cdot e^{0.12(\pm0.03)\text{T}}$$  (1)

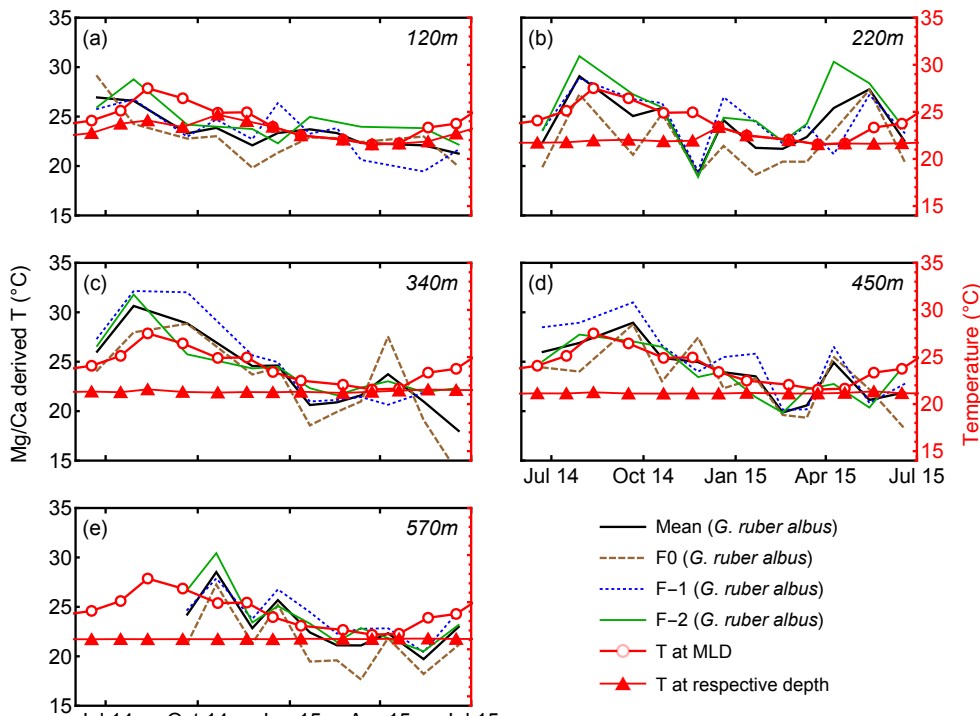


Figure 10: *G. ruber albus* Mg/Ca-derived temperatures versus measured temperatures (red).
The calculated temperatures were derived from Eq. 1 for individual chambers. See also Levy
et al. (2023).

Inter-chamber variability (ICV) has been shown to affect the local Mg/Ca temperature
calibration (Eq. 1) of *G. ruber albus* (Levy et al., 2023; Fig. 10). Generally, Mg/Ca-derived
temperatures from chambers F-1 and F-2 correspond closely with mixed layer depth (MLD)
temperatures. However, beneath the photic zone, Mg/Ca of all three chambers F0, F-1, and F-
2 of the *G. ruber albus* specimens exhibit poor fits with measured temperatures (Fig. 10). Given
that *G. ruber albus* calcifies its shell in the photic zone (Schiebel and Hemleben, 2017), these
findings support the use of Mg/Ca as a paleothermometer (Nürnberg et al., 1996). The Mg/Ca-





derived temperatures from chamber F0 show lower Mg/Ca temperatures of the MLD than
chambers F-1 and F-2 (Fig. 10). Although Mg/Ca data from chambers F-1 and F-2 appear
suitable for reconstructing temperatures and demonstrate agreement with MLD temperature
trends, the high ICV in *G. ruber albus* is evidently too great to accurately reflect ambient
temperatures using this calibration. Therefore, and based on these new observations, we
suggest that optimal Mg/Ca-temperature calibration (Eq. 1) should be based on the pooled
mean of the F-1 and F-2 chambers at all depths.

4.2 B/Ca as a proxy for *p*H
B/Ca in some PF species has been suggested to be a proxy for *p*H (Yu et al., 2007; Allen et al.,
2011). Comparing chamber B/Ca of both *G. ruber albus* and *T. clarkei* (both 'big' and
'encrusted') alongside *p*H at various water column depths in the GOA reveals contrasting
results. While B/Ca in *G. ruber albus* exhibits seasonality (Fig. 6), with lower values during
winter months, it does not appear to be consistent with the *p*H of respective water depth nor
the MLD (Fig. 11). This inconsistency suggests that B/Ca in *G. ruber albus* from the GOA is
not a reliable recorder of ambient water *p*H. Similarly, Henehan et al. (2015) and Naik & Naidu
(2014) reported that B/Ca of open ocean core-top samples and down-core sediment samples do
not display a *p*H relationship.
Alternatively, B/Ca in *G. ruber albus* may be sensitive to salinity and micro-environments
produced by PF symbionts with *p*H levels which are distinct from the ambient water column.
Culture experiments have shown that B/Ca is affected by salinity and increases with increasing
salinity (Allen et al., 2012). However, only small salinity changes occur in the GOA (Fig. 1),
which argue against a strong B/Ca-salinity relationship that would result in a B/Ca seasonal
trends. It was suggested that photo-symbionts such as dinoflagellates in *G. ruber albus* create
micro-environments with *p*H levels, which are distinct from ambient seawater, to accommodate
for their photosynthetic activity, and indicate that B/Ca is more affected by *p*H in those micro-
environments than the water column *p*H (Hönisch et al., 2021; Babila et al., 2014). An
additional observation for the *G. ruber albus* B/Ca values is that they are relatively high in
comparison to values from other studies. The relatively high salinity in the GOA (~41),
combined with the photosymbiont activity in *G. ruber albus* may explain the elevated B/Ca
values (Henehan et al., 2015; Hönisch et al., 2021; Babila et al., 2014).
In contrast to *G. ruber albus*, B/Ca in the photosymbiont barren *T. clarkei* may indeed record
the changes in *p*H (Fig. 11) of seawater at its ambient dwelling depth, possibly shifting between
the deeper water column depth horizons where *p*H changes are evident. Indeed, based on the




fluxes of *T. clarkei* (Chernihovsky et al., 2018; Fig. 2), the B/Ca of *T. clarkei* in the sediment
record likely represent the *p*H beneath the thermocline and within the deep-water column
horizons for specimens that lived from early winter through spring. In particular, *p*H at 340 m
appears to correlate with the B/Ca trends of *T. clarkei* types. For B/Ca-*p*H calibrations utilizing
the pooled mean of data from the chambers F-1, F-2, and F-3 is recommended, while excluding
the    F0    and    F-4    chambers    where    more    ICV    is    apparent    (Fig.    11).

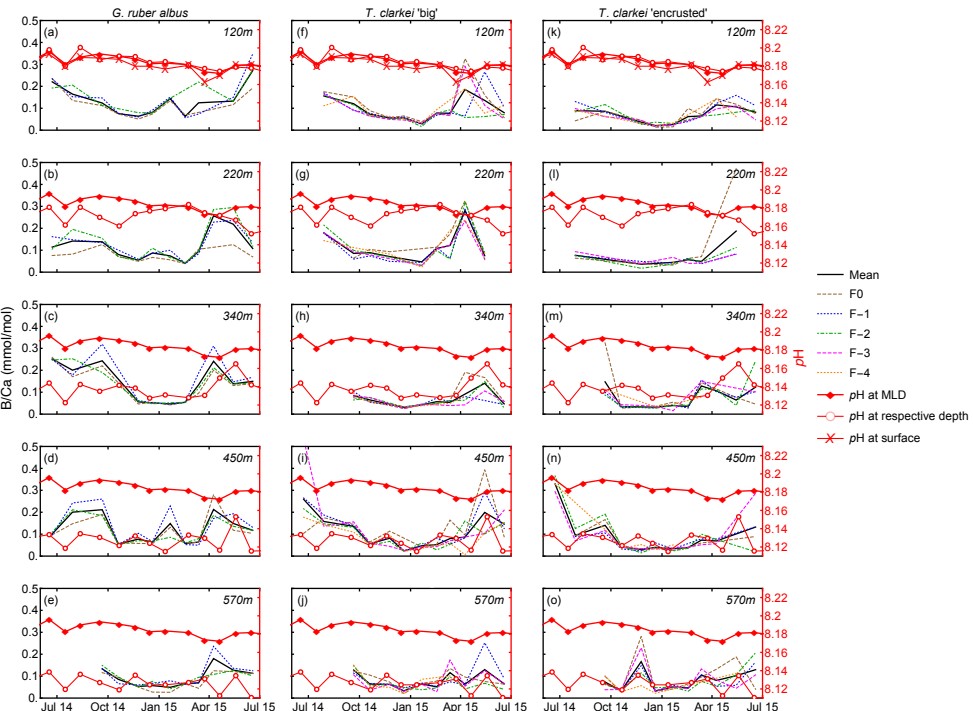

Figure 11: Single chamber B/Ca and in situ *p*H measured at MLD depth (empty red circles)
and 120 m water depth for *G. ruber albus*, *T. clarkei* 'big' and *T. clarkei* 'encrusted'.

4.3  Na/Ca as a proxy for salinity and Ba/Ca as an indicator for productivity

Cultured individuals and samples from the surface Caribbean and the Gulf of Guinea of live *T.*
*sacculifer* indicate that Na/Ca can be used as a proxy for salinity, without temperature
dependence, however, a species-specific calibration might be required (Bertlich et al., 2018).
Despite the high variability of PF Na/Ca values in the GOA during water column mixing (Fig.
7), salinity remains high and relatively constant, ranging between 40.4-40.7. Consequently, a
local Na/Ca-salinity calibration shows no significant relationship for any of the three PF
morphotypes.




Na/Ca values in PF from the GOA are notably higher compared to other regions. Gray *et*
*al.* (2023) explored the relationship between Na/Ca and salinity in *G. ruber albus* collected
from sediment traps, plankton tows, culture samples, and core top samples, contributing to the
ongoing discussions regarding the reliability of Na/Ca as a proxy for salinity in both planktic
and benthic foraminifera (Allen et al., 2016; Geerken et al., 2018; Mezger et al., 2016, 2018;
Gray et al., 2023, and references therein). They concluded that the measurement method (i.e.,
'solution' ICP-MS vs. LA-ICP-MS) influences the values of Na/Ca and in turn the relationship
with salinity, i.e., weak in solution-based compared to significant in laser ablation-based, at
salinity over 36.
Comparing Na/Ca of *G. ruber albus* from the shallowest sediment trap (120 m) in the GOA
with the Na/Ca of *G. ruber albus* plankton tows-samples from the GOA deployed and collected
in January 2010 and October 2013 (Gray et al., 2023), both measured using LA-ICP-MS,
generally reveals similar results, excluding the high-value excursions observed in some single
chamber measurements (Fig. 7). *Turborotalita clarkei* in the GOA exhibits elevated Na/Ca
values in both 'big' and 'encrusted' compared to *G. ruber albus*. Unlike *G. ruber albus*, there
is relatively higher variability between water depths as well as significantly higher values in
March, April, and May associated with water column mixing (Fig. 7). During these mixing
events, nutrient-rich, high salinity (∼40.7) water ascend upward. Therefore, the Na/Ca of *T.*
*clarkei* may serve as a proxy for water column stability, i.e., stratification vs. mixing.
The Ba/Ca in the three morpho-species show a relatively strong correlation with Na/Ca
(0.74 and 0.82 in *T. clarkei* 'big' and 'encrusted' respectively, and 0.54 in *G. ruber albus*, the
second highest ratio and exceeded only by the 0.57 of Pb/Ca). Ba/Ca is presumably unaffected
by temperature, salinity, and *p*H (Hönisch et al., 2011). In non-spinose species, Ba/Ca typically
shows positive relationships with productivity and potentially can be used as an indicator of
river run-off (Fritz-Endres et al., 2022; Hönisch et al., 2011; Weldeab et al., 2014). Although
floods in the catchment area of the GOA are brief and occur only few times each year (Katz et
al., 2015), significant Ba/Ca perturbations during water column mixing may reflect nutrient-
rich water admixing to the surface water (Fig. 8).

5.  Discussion:

5.1 Temporal and vertical dynamics of element/Ca in the GOA
Trace element incorporation into the calcium carbonate shells of planktic foraminifera during
calcification is controlled by environmental and ecological factors in the water column such as




temperature, salinity, *p*H, the carbonate system, dust and terrigenous inputs, as well as whether
a species harbor photosymbionts (Schiebel & Hemleben, 2017; and others). Shells of *G. ruber*
*albus*, *T. clarkei* 'big' and *T. clarkei* 'encrusted' from the GOA show species-specific behavior
and offer new insights into how these species respond to the vertical and temporal variations
in the water column. For most elements, the smaller *T. clarkei* specimens display higher trace
element ratios than the larger *G. ruber albus*, suggesting more efficient trace element
incorporation to the shell or implying that its habitat deeper in the water column has conditions
which result in higher trace element incorporation (Fig. 3). Some element ratios such as Mg/Ca,
Sr/Ca, B/Ca, Na/Ca (for *G. ruber albus*) and Ba/Ca for both *G. ruber albus* and *T. clarkei*
'encrusted', show overlap between specimens from the water column and from core-tops (Fig.
3), confirming the robustness of downcore-based records allowing to further consider these
element/Ca recorders of the water column as paleo-proxies.
While water depth likely influences element/Ca through variations in physical and
chemical conditions, the observed inter-chamber variability (ICV) and element/Ca differences
between species cannot be attributed to any single environmental parameter. Nonetheless,
elements such as Al/Ca, Ti/Ca, Mn/Ca, and Fe/Ca for all species, and Mg/Ca, Sr/Ca, Na/Ca,
and Ba/Ca for *G. ruber albus* alone, demonstrate consistent behavior across the water column,
suggesting that depth-related factors do not significantly alter calcification mechanisms. This
supports the use of pooled mean values for specimens over multiple sediment traps spread over
depths (Levy et al., 2023). Interestingly, most element/Ca peak during water column mixing in
March-April 2015 for all three morphotypes analyzed here, accompanied by larger ICV (Figs.
4-8). Mg/Ca in *G. ruber albus* and Sr/Ca in all three species show less pronounced excursions,
while other trace element ratios (e.g., Co/Ca, U/Ca) exhibit more variability and more extreme
values (Figs. S5 and S10). These observations can reflect: i) primary calcite structure
alterations driven by environmental shifts and life cycle changes, ii) secondary mineralization
(e.g., barite, Amorphous Calcium Carbonate, ACC) (Torres et al., 2010; Evans et al., 2020 and
references therein), and iii) fluid inclusions within the shell structure (Gray et al., 2023).
All of these relationships do possibly concern the ontogenetic PF calcite, since SEM
imaging of GOA specimens did not reveal secondary minerals or overgrowth on shell calcite
(Levy et al., 2023). Moreover, the enrichment of multiple trace elements across species
suggests that secondary minerals are unlikely to be responsible for these trends. Discrepancies
between Na/Ca in plankton tow versus core-top samples in the Red Sea (Mezger et al., 2018),
as well as higher Na/Ca values measured by LA-ICP-MS compared to solution ICP-MS, have
been linked to early diagenesis of Na-enriched phases like spines, ACC, or fluid inclusions





(Gray et al., 2023). However, spines and ACC were ruled out for GOA samples, as all of the
specimens had lost their spines before analysis and ACC was not detected via SEM. Given that
most element/Ca in GOA shells are elevated relative to PF data from elsewhere, fluid inclusions
may be a contributing factor (Gray et al., 2023). However, more research is required to
investigate whether fluid inclusions are evident in PF shells from the GOA. In the absence of
fluid inclusions, environmental changes, particularly during water column mixing, are
considered to be the primary drivers of the observed trace element/Ca enrichments in the GOA.

5.2  Water column and sediment signal correlation: Implications to Paleoceanographic

studies

Several element ratios (e.g., Al/Ca, Ti/Ca, Mn/Ca, Fe/Ca, Nd/Ca, U/Ca, Co/Ca, and Th/Ca)
exhibit discrepancies between water column and core-top specimens (Fig. 3). Some, like
Co/Ca, have lower values in surface sediment than the water column, while others, like Fe/Ca
show higher values. Differences between sediment trap samples and core-top samples may
stem from differential diagenetic processes that affect element/Ca in specimens taken from the
water column and the sea floor. For example, diagenetic processes can lead to Mn accumulation
and higher Mn/Ca in PF from the core top (McKenzie, 1980; Steiner et al., 2017). Conversely,
core-top PF samples may show lower ratios due to the release of these metals into pore water
over time (e.g., Co/Ca, Fig. 3i). This release can alter the elemental composition, potentially
skewing paleoenvironmental reconstructions. Understanding these processes is crucial for
accurately interpreting geochemical data from both sample types.

Despite the offsets of Al/Ca and Ti/Ca between core top and water column specimens, they

nevertheless may be utilized to trace the origins of terrigenous inputs and identify periods of
dust deposition in the geological record (Torfstein et al., 2017; Martinez-Garcia et al., 2011).
Our data reveal significant seasonal excursions in Al/Ca and may demonstrate the use of Al/Ca
and Ti/Ca in PF tests as proxies for dust or terrigenous input to the ocean (Fig. S3).

Core top element/Ca values that fall within the same range of values of the sediment trap

specimens (Mg/Ca, Sr/Ca, B/Ca, Na/Ca, and Ba/Ca; Fig. 3) suggest that they could reflect
water column conditions. The high temporal variability in many of these element/Ca data,
together with the varying PF population dynamics throughout the year (Fig. 2) may be
considered when approaching PF from sediment cores. Seasonal trends in element/Ca are often
obscured by the spring mixing event. However, exceptions to this are observed in Mg/Ca for
*G. ruber albus* (Fig. 4; Levy et al., 2023) and B/Ca for *T. clarkei* (Fig. 6), where clear seasonal
patterns emerge. A key limitation of reconstructing past environments from element/Ca in PF



shells is the challenge of disentangling seasonal effects from other more episodic
environmental signals. However, by identifying water column mixing events through positive
element/Ca excursions and elevated ICV, which are evident across all three species (Figs. 4-
8), it may be possible to identify the time intervals over which environmental changes are
reconstructed. This could allow for more accurate reconstructions of shifts in temperature,
carbonate chemistry, and nutrient availability during specific mixing events, improving our
understanding of past ocean conditions.
5.3 Regional comparison of geochemical conditions and PF element/Ca
The Mg/Ca, Al/Ca, and Na/Ca in PF from the GOA generally exceed those reported from other
regions (Fig. 12b – 14e). Sr/Ca values, while reaching up to 2.2 mmol/mol during spring, have
an average of 1.5 mmol/mol, consistent with previous studies (Fig. 12c; Kisakürek et al., 2008;
Cleroux et al., 2008; Elderfield et al., 2002; Brown & Elderfield, 1996; Dissard et al., 2021).
The high Mg/Ca range in the GOA versus typical open-ocean levels (0.5-5 μmol/mol) is
attributed to elevated salinity (~ 41 compared to mean ocean values of 34.7), which is also
evident by the high Na/Ca. The high Al/Ca values and their large variation may be attributed
to the close proximity of GOA to terrestrial input. Ba/Ca in the GOA are significantly higher
than the values reported in prior studies from Atlantic Ocean core samples and culture
experiments (Hönisch et al., 2011; Lea & Boyle, 1991), representing a roughly ten-fold
difference. These discrepancies likely stem from two factors: (1) higher salinity in the GOA
increases the availability of cations and trace element incorporation into foraminifera shells,
and (2) higher-resolution measurements here which reveal chamber-specific elemental ratios,
where early chambers (F-1 and F-2) exhibit higher values than final chambers, leading to more
accurate, chamber-level data compared to bulk measurements. Combined, these factors explain
the elevated values relative to global reports.




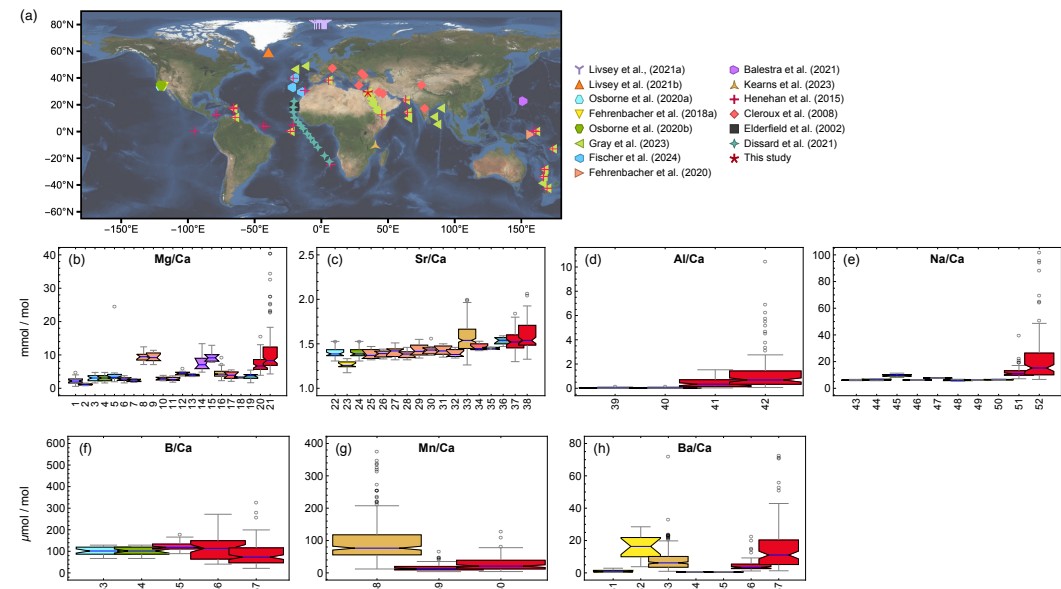


Figure 12. Global comparison of major and trace element-to-calcium ratios. (a) sample global map, (b) Mg/Ca of *N. pachyderma*, *G. bulloides*, *G. ruber white*, *N. dutertrei*, *O. universa*, *P. obliquiloculata*, *T. sacculifer* and *T. clarkei* derived from various sources (plankton tows/nets, sediment traps, cores) and measured by Laser Ablation (LA)-ICP-MS, solution-ICP-MS (SOL) and Electron micro-probe analyses (EPMA). (c) Sr/Ca of *G. bulloides*, *G. ruber white*, *N. dutertrei*, *O. universa*, *P. obliquiloculata*, *T. sacculifer* and *T. clarkei* derived from various sources (plankton tows/nets, sediment traps, cores) and measured by LA-ICP-MS and solution-ICP-MS. (d) Al/Ca of *G. bulloides*, *G. ruber white* and *T. clarkei* derived from sediment traps and measured by LA-ICP-MS. (e) Na/Ca of *G. ruber white* and *T. clarkei* from various sources (plankton tows/nets, sediment traps, cores and cultured samples) and measured by LA-ICP-MS and solution-ICP-MS. (f) B/Ca of *G. bulloides*, *G. ruber white* and *T. clarkei* derived from sediment traps and measured by LA-ICP-MS. (g) Mn/Ca of *G. ruber white* and *T. clarkei* derived from cores and sediment traps and measured by LA-ICP-MS. (h) Ba/Ca of *G. ruber white*, *N. dutertrei* and *T. clarkei* derived from various sources (plankton tows/nets, sediment traps, cores and cultured samples) and measured by LA-ICP-MS. See table 1 for detailed description of methods.






| # | Element/Ca | Reference | Species | Collecting method | Measuring method |
|---|---|---|---|---|---|
| 1 | Mg/Ca | Livsey et al. (2021a) | *N. pachyderma* | Plankton tows / nets | LA |
| 2 | Mg/Ca | Livsey et al. (2021b) | *N. pachyderma* | Sediment traps | LA |
| 3 | Mg/Ca | Osborne et al. (2020) | *G. bulloides* | Sediment trap | LA |
| 4 | Mg/Ca | Osborne et al. (2020b) | *G. bulloides* | Sediment trap | LA |
| 5 | Mg/Ca | Fischer et al. (2024) | *G. ruber* | Plankton tows / nets | LA |
| 6 | Mg/Ca | Fehrenbacher et al. (2020) | *N. dutertrei* | Core | LA |
| 7 | Mg/Ca | Fehrenbacher et al. 2020 | *N. dutertrei* | Core | SOL |
| 8 | Mg/Ca | Fehrenbacher et al. (2020) | *O. universa* | Core | LA |
| 9 | Mg/Ca | Fehrenbacher et al. (2020) | *O. universa* | Core | SOL |
| 10 | Mg/Ca | Fehrenbacher et al. (2020) | *P. obliquiloculata* | Core | LA |
| 11 | Mg/Ca | Fehrenbacher et al. (2020) | *P. obliquiloculata* | Core | SOL |
| 12 | Mg/Ca | Fehrenbacher et al. (2020) | *T. sacculifer* | Core | LA |
| 13 | Mg/Ca | Fehrenbacher et al. (2020) | *T. sacculifer* | Core | SOL |
| 14 | Mg/Ca | Balestra et al. (2021) | *O. universa* | Plankton tows / nets | EPMA |
| 15 | Mg/Ca | Balestra et al. (2022) | *O. universa* | Plankton tows / nets | EPMA |
| 16 | Mg/Ca | Kearns et al. (2023) | *G. ruber* | Core | LA |





| 17 | Mg/Ca | Cleroux et al. (2008) | *G. ruber* | Core | SOL |
|---|---|---|---|---|---|
| 18 | Mg/Ca | Elderfield et al. (2002) | *G. ruber* | Core | SOL |
| 19 | Mg/Ca | Dissard et al. (2021) | *T. sacculifer* | Plankton tows / nets | LA |
| 20 | Mg/Ca | This study | *G. ruber* | Sediment trap | LA |
| 21 | Mg/Ca | This study | *T. clarkei* | Sediment trap | LA |
| 22 | Sr/Ca | Osborne et al. (2020) | *G. Bulloides* | Sediment trap | LA |
| 23 | Sr/Ca | Fehrenbacher et al. (2018a) | *N. dutertrei* | Plankton tows / nets | LA |
| 24 | Sr/Ca | Osborne et al. (2020b) | *G. bulloides* | Sediment trap | LA |
| 25 | Sr/Ca | Fehrenbacher et al. (2020) | *N. dutertrei* | Core | LA |
| 26 | Sr/Ca | Fehrenbacher et al. (2020) | *N. dutertrei* | Core | SOL |
| 27 | Sr/Ca | Fehrenbacher et al. (2020) | *O. universa* | Core | LA |
| 28 | Sr/Ca | Fehrenbacher et al. (2020) | *O. universa* | Core | SOL |
| 29 | Sr/Ca | Fehrenbacher et al. (2020) | *P. obliquiloculata* | Core | LA |
| 30 | Sr/Ca | Fehrenbacher et al. (2020) | *P. obliquiloculata* | Core | SOL |
| 31 | Sr/Ca | Fehrenbacher et al. (2020) | *T. sacculifer* | Core | LA |
| 32 | Sr/Ca | Fehrenbacher et al. (2020) | *T. sacculifer* | Core | SOL |
| 33 | Sr/Ca | Kearns et al. (2023) | *G. ruber* | Core | LA |
| 34 | Sr/Ca | Cleroux et al. (2008) | *G. ruber* | Core | SOL |



| 35 | Sr/Ca | Elderfield et al. (2002) | *G. ruber* | Core | SOL |
|---|---|---|---|---|---|
| 36 | Sr/Ca | Dissard et al. (2021) | *T. sacculifer* | Plankton tows / nets | LA |
| 37 | Sr/Ca | This study | *G. ruber* | Sediment trap | LA |
| 38 | Sr/Ca | This study | *T. clarkei* | Sediment trap | LA |
| 39 | Al/Ca | Osborne et al. (2020) | *G. Bulloides* | Sediment trap | LA |
| 40 | Al/Ca | Osborne et al. (2020b) | *G. bulloides* | Sediment trap | LA |
| 41 | Al/Ca | This study | *G. ruber* | Sediment trap | LA |
| 42 | Al/Ca | This study | *T. clarkei* | Sediment trap | LA |
| 43 | Na/Ca | Gray et al. (2023) | *G. ruber* | Core | SOL |
| 44 | Na/Ca | Gray et al. (2023) | *G. ruber* | Cultured | SOL |
| 45 | Na/Ca | Gray et al. (2023) | *G. ruber* | Plankton tows / nets | LA |
| 46 | Na/Ca | Gray et al. (2023) | *G. ruber* | Plankton tows / nets | SOL |
| 47 | Na/Ca | Gray et al. (2023) | *G. ruber* | Sediment trap | LA |
| 48 | Na/Ca | Gray et al. (2023) | *G. ruber* | Sediment trap | SOL |
| 49 | Na/Ca | Gray et al. (2023) | *G. ruber mixed* | Core | SOL |
| 50 | Na/Ca | Gray et al. (2023) | *G. ruber sl* | Core | SOL |
| 51 | Na/Ca | This study | *G. ruber* | Sediment trap | LA |
| 52 | Na/Ca | This study | *T. clarkei* | Sediment trap | LA |




| 53 | B/Ca | Osborne et al. (2020) | *G. Bulloides* | Sediment trap | LA |
|---|---|---|---|---|---|
| 54 | B/Ca | Osborne et al. (2020b) | *G. Bulloides* | Sediment trap | LA |
| 55 | B/Ca | Henehan et al. (2015) | *G. ruber* | Core | SOL |
| 56 | B/Ca | This study | *G. ruber* | Sediment trap | LA |
| 57 | B/Ca | This study | *T. clarkei* | Sediment trap | LA |
| 58 | Mn/Ca | Kearns et al. (2023) | *G. ruber* | Core | LA |
| 59 | Mn/Ca | This study | *G. ruber* | Sediment trap | LA |
| 60 | Mn/Ca | This study | *T. clarkei* | Sediment trap | LA |
| 61 | Ba/Ca | Fehrenbacher et al. (2018a) | *N. dutertrei* | Cultured | LA |
| 62 | Ba/Ca | Fehrenbacher et al. (2018a) | *N. dutertrei* | Plankton tows / nets | LA |
| 63 | Ba/Ca | Kearns et al. (2023) | *G. ruber* | Core | LA |
| 64 | Ba/Ca | Hönisch et al. (2011) | *G. bulloides* | Cultured | SOL |
| 65 | Ba/Ca | Hönisch et al. (2011) | *O. universa* | Cultured | SOL |
| 66 | Ba/Ca | This study | *G. ruber* | Sediment trap | LA |
| 67 | Ba/Ca | This study | *T. clarkei* | Sediment trap | LA |


Table 1: detailed description of the different species, measurement methods and sample
origin used for the compilation in figure 12. LA stands for Laser Ablation (LA)-ICP-MS,
SOL is solution-ICP-MS and EPMA is Electron micro-probe analyses.

6. Summary and conclusions:





We investigated the effects of inter-chamber variability on the proxy systematics in the hyper
saline oligotrophic GOA using single chamber LA ICP-MS analysis measured on two flux-
dominating planktic foraminifer (PF) species *G. ruber albus* and *T. clarkei* with its two
phenotypes 'big' and 'encrusted'. We observed how element/Ca varies in PF chambers as a
function of environmental changes in order to then be used as proxies for past oceanic and
climatic reconstruction. The results show that some element/Ca exhibit temporal and seasonal
variations related to environmental conditions in the water column such as Mg/Ca in *G. ruber*
*albus* as a temperature proxy, and B/Ca in *T. clarkei* as a proxy of *p*H. Although other
element/Ca values display more limited variability (e.g., Na/Ca) they may still be of use as
paleo-proxies when combined in global calibration studies.
Water column mixing has been shown to have a significant effect of element/Ca positive
excursions in the analyzed *G. ruber albus*, and two *T. clarkei* morphotypes, which may limit
the use of some element ratios as proxies, or alternatively, be used as a proxy for water column
mixing. Generally, pooled-mean values of element/Ca in the PF tests in the GOA are species-
specific and element-specific, and are elevated compared to other regions (e.g., Mg/Ca, Al/Ca,
Na/Ca). However, the final chamber F0 is different in comparison to the preceding chambers
F-1 and F-2, suggesting that the element composition of F0 may be biased and unreliable in
terms of recording environmental conditions.
Our findings indicate that high-resolution analytical techniques, such as LA ICP-MS
enable studying single chamber compositions and variations. Although pooled mean values of
specimens over various water depths are recommended for their incorporation as proxies, ICV
can also be used as a tracer of environmental factors. Exploring different biochemical or
physiological mechanisms which are responsible for the element/Ca variations between species
and chambers are critical to shed light on how element/Ca are incorporated to the PF shells.
Despite these limitations, the results provide valuable insights into the complex behavior of
element/Ca in PF shells.
Data availability
Tabular supplementary data generated in this study can be found at PANGAEA (DOI: *will be*
*added following acceptance*).

Author contributions



NL, AT, and RS designed the study; NL, BS, UW, and KPJ, performed the measurements;
NL, NC, AT, and RS analyzed the data; NL, RS and AT wrote the manuscript draft; NL, RS,
AT and GH reviewed and edited the manuscript.

The authors declare that they have no conflict of interest.

Acknowledgments
We wish to acknowledge the IUI marine crew and B. Yarden for their assistance in field work
and sample handling. The National Monitoring Program are thanked for their support and
sharing results and E. Levy for fruitful discussions. We are thankful for the three anonymous
reviewers whom their comments significantly improved this manuscript. This work was
supported by Israel Science Foundation grant 834/19 (to AT), a Minerva PhD Fellowship
stipend (to NL) and a scholarship from the Advance School for Environmental Studies, HUJI
(to NL).




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
