# Peer review of "Monthly element/Ca trends and inter chamber variability in two planktic Foraminifera"

_EGUsphere, 2025_

## Author Comment (AC1)

Dear Takashi,

Thank you very much for investing your time and effort in reviewing our paper. We will address your major points and afterwards the minor suggestions in an itemized list:

**Item 1: Na/Ca spikes**

**The reason for Na/Ca spike cannot be explained by the increase of Na concentration in seawater. Foraminiferal Na/Ca is also an indicator of salinity. If we try to explain the large Na/Ca variation as in Fig. 7 by seawater Na/Ca variation, we have to suppose an event in which salinity increases by two digits scale. However, from the desert in the hinterland, even if minerals are input and somewhat dissolved, it is hard to consider an impact on the salinity at the scale of digit change. Also, from the analysis of T. clarkei in Fig. 9b, c, Na shows positive correlation with Mg, Al, Ti, Mn, Fe. This may suggest that T. clarkei has the property to incorporate sinking particles containing Na, such as Albite/Na-feldspar and Plagioclase, on/into the shell during calcification. I guess the possibility of particle trap on the shell. This is possibly indicating a new role of foraminifera as "fossils that trapped sinking particles" in addition to being environmental proxies. From Fig. 7(b), there is a possibility that the same phenomenon is happening in G. ruber. If this is because the study area has an arid region in the background and seasonally sinking particles become extremely abundant in seawater, this does not affect the soundness of proxies using planktonic foraminifera from other regions. Also, if we can monitor some elements like Ti, Na or Si to check whether the proxy is working normally, the soundness of the environmental proxy in the study area is also kept, while the role as a catcher of sinking particles itself will emerge.**

**From the perspective of calcification mechanisms, the calcifying fluid is to some extent isolated from ambient seawater, making the direct incorporation of external particles unlikely. However, it is possible that particles adhering to the shell surface become enclosed when a new chamber is formed over them. Although unpublished, in my own experience this reviewer has observed cases where diatom frustules were incorporated into the interior of the shell. In other words, the incorporation of foreign material into the shell interior can indeed occur. While such occurrences have generally been rare enough to go unnoticed, this study might find by the possibility that in certain seasons in this particular region, such incorporation might happen more frequently.**

**To verify whether the Na/Ca spikes originate from calcite itself or from the incorporation of external mineral particles, it is necessary to conduct some form of direct check. For example, confirming the amount and seasonal changes of sinking particles in the study area, and performing SEM observations or XRD analysis of the shells, would allow you to determine whether mineral-like foreign materials are present inside the calcite. Alternatively, by examining the depth-resolved elemental profiles obtained from the authors' LA-ICP-MS analyses, it should be possible to determine whether the influence of external particles extends throughout the entire calcite structure or is confined to specific locations.**

**Establishing this point is essential for assessing the reliability of Na/Ca as a proxy in this environment, and solving it would also strengthen the discussion of Fig. 12.**

*Reply 1: Thank you very much for the thorough and comprehensive points regarding the Na/Ca 'spikes' in the three species from the hyper-saline Gulf of Aqaba.*
*Considering the new role of foraminifera as "fossils that trapped sinking particles" in addition to being environmental proxies: we have reexamined the relationships of Na/Ca with Al/Ca, Ti/Ca, Mn/Ca and Fe/Ca in the three species in a 'Spearman correlation' matrix and we see a weak correlation in G. ruber albus and a strong correlation in T. clarkei 'big' (slightly weaker in T. clarkei 'encrusted') as well as a strong correlation to the environmental variable MLD in T. clarkei 'big'. Potentially, this may suggest that during water column mixing some foreign particles are incorporated into T. clarkei shells (suspended particles?). The close proximity of station A in the GOA to the nearby hyper-arid desert land may be an important aspect leading to this phenomenon and we will check for data of amount and seasonal changes of sinking particles in the study area as you suggested but is beyond the scope of this manuscript (we will include it in our future paper on T. clarkei: Levy et al., in prep). In the meantime, we show here SEM images of three samples (i.e., three individuals) in which we have not observed any particles enclosed in their shells. We attach here several SEM images of T. clarkei which will be published in a future paper (Levy et al., in prep) (Figure 1 below):*

[Figure]

*Figure 1: SEM images of T. clarkei 'big' and T. clarkei 'encrusted'.*

*Regarding checking the Na ablation profiles: we calculated the average element-to-calcium ratio from the spot derived LA-ICP-MS count data from just after the start of the ablation peak until where we believe the end of calcite is based on Mg/Ca ablation profile. This time average should amount to the portion of the ablation signal that represents the stable internal material of the shell, rather than the noisy beginnings or ends of the ablation event*

*(see the Figure 2 below for Na23 count signal and raw Na/Ca of G. ruber at 220m depth April 2015 as an example).*

[Figure]

*Figure 2: A timeseries plot of (a) Na23 counts and (b) respective Na/Ca for F0, F-1, and F-2 in G. ruber taken from 220 m sediment trap in April 2015. Plot markers show the data used for calculating Na/Ca averages.*

*In some cases, we found that both the element signals and the middle of the ablation peak were very noisy and elevated relative to the other chambers (see Figure 3: F0 in G. ruber at 120m depth April 2015 as an example). In these cases, we could not accurately ascertain the El/Ca ratios and the values are not reported in the supplementary table S4). Thanks to your point raised we rechecked the Na/Ca ablation profiles of some of the unusually high Na/Ca 'spike' measurements and found elevated Na/Ca intensities towards the end of the ablation event in a measurement which calculated to Na/Ca= 64.1mmol/mol - the highest Na/Ca value in the G. ruber dataset (see Figure 3 here: F-2 in G. ruber at 120m depth April 2015). It is important to note that for this measurement (F-2 in G. ruber at 120m depth April 2015), the other El/Ca ratios did not show unusually high values (Mg/Ca, B/Ca, Sr/Ca, Al/Ca, Ti/Ca,*

*Mn/Ca, Fe/Ca, we did not measure Si/Ca) which suggests that the ablation still measured the chamber wall. However, an additional 'foreign' particle would indeed be a possibility, especially considering the unusual F0 measurement in the same specimen.*

[Figure]

*Figure 3: A timeseries plot of (a) Na23 counts and (b) respective Na/Ca for F0, F-1, and F-2 in G. ruber taken from 120 m sediment trap in April 2015. Plot markers show the data used for calculating Na/Ca averages.*

**Item 2:** **Final chamber (F0) composition**

**Regarding the idea that F0 (final chamber) element composition does not reflect the environment, I think there are both opinions, but to deny it here needs a little more basis. For example, Sadekov et al. (2009: https://doi.org/10.1029/2008PA001664) concluded that Mg/Ca of the final chamber has the highest correlation with temperature, and Hupp and Fehrenbacher (2024: https://doi.org/10.61551/gsjfr.54.4.355) also did not point out problems in analyzing the final chamber. There are other similar studies. Especially Mg/Ca has**

**many cases that respond rather straightforward to temperature changes, so I would be more convinced if you point out that calcification temperature (depth) is different from the assumption.**

*Reply 2: Yes, we agree that F0 does reflect the environmental conditions but records El/Ca slightly differently than the other chambers. In the revised manuscript we clarified this and add the mentioned references (Sadekov et al., 2009; and Hupp and Fehrenbacher, 2024), respectively (section 4.3 in the revised manuscript (lines 638-651).*

**Item 3: Small number of individuals**

**I appreciate again the accumulation of efforts that you analyze three categories of foraminifera at each depth every month, which is very ambitious. However, the small number of individuals in each population is obvious. ICV is discussed, but I do not find quantitative treatment of inter-individual variability or pooled mean value. As the basis to say that discussion is possible with few individuals, could you add, in addition to pooled mean value, statistical indices showing the magnitude of variation among individuals (for example: standard deviation, coefficient of variation) or excuses from previous studies which state that comparing by pooled mean value for inter-individual variation is no problem?**

*Reply 3:*
*In the revised manuscript we now report standard deviations (SD) in each specimen (individual foram) as a measure of Inter-chamber variability in the results section 3.1 for the reported El/Ca discussed respectively (and include a supplementary table S4). We included statistical analyses of the SD in the new 'Spearman correlation matrix' which reveals environmental parameters, such as MLD, correlative to ICV in some species (i.e. T. clarkei 'big') (supplementary figure S12 and, also in Figure 1 in the reply to reviewer 1). The pooled means and SD for each chamber in all specimens taken at each given time interval as a function of time is shown, as well as the total pooled mean (lower timeseries panels in the revised figures 3-7) (see also in Figure 2 in the reply to reviewer 1). We additionally include biplot's summarizing Redundancy analysis (RDA) per species and El/Ca to examine the relationship between depth El/Ca and environmental parameters. Furthermore, we have appended the 'Spearman correlation' matrix with environmental parameters (MLD, T, S, pH; Figure 9 in the revised manuscript).*

**Item 4: Minor points**

    a) **Fig. 2 appears quite late in the text. You forget to refer to Fig. 2 somewhere in the first half.**
    b) **In the text final chamber is written as F0, but in Fig. 1 it is F-0. Please unify.**
    c) **In Fig. 4 etc., please indicate MLD also in the legend.**
    d) **Comparing the environmental figure in Fig. 1 with Fig. 3–8, the horizontal axis is shifted. Please fix, and please also write what the horizontal axis represents.**
    e) **In Fig. 9, elements are slightly misaligned with the columns. For example, U is completely showing the result of the previous element for both vertical and horizontal axes.**

*Reply 4: Thank you for the following comments.*

a) *We now refer to figure 2 much earlier in the introduction –line 116 in the revised manuscript.*
b) *F-0 in Figure 1 is now changed to F0.*
c) *Figures 4-8 moved and now titled 3-7. They all have a top panel per species of the MLD for better visualization. The MLD is now also present in the legend.*
d) *Regarding the environmental parameters in Figure 1 we have added a title to the x-axis but have opt to leave the horizontal axis longer than in figures 3-7 as we find the 'wide' perspective is more suitable.*
e) *Figure 9 has now changed – elements are now aligned with the columns; more parameters are available (environmental parameters) and we have changed the axis to be from -1 to 1 (previously it wrongly displayed as 0 – 1 which altered the color gradients).*

Sincerely,

Noy Levy on behalf of all co-authors

---

## Author Comment (AC2)

Response to the referee: Lennart de Nooijer

Dear Lennart,

Thank you very much for your very much appreciated comments and suggestions in your review of our paper. We have addressed all of them. Please see the itemized list below (the reviewer comments in bold):

Item 1: **In summary: not just the means, but the full single-chamber El/Ca should be shown and (statistically) analyzed. Now, only the standard error is shown (figure 3, although very difficult to distinguish). There are multiple questions that the authors could answer:**
   a- **what exactly is the between-chamber variability in El/Ca and**
   b- **how does this relate to the chamber number?**
   c- **Does that change with time?**
   d- **Is it similar between depths and is it similar for the different elements? If there are differences, are they significant?**

*Reply 1:* *To address the lack of statistical analysis and information in the paper, we implemented a number of changes: 1) We've now reported standard deviations (SDs) in each specimen (individual foram) as a measure of inter-chamber variability for each discussed El/Ca (results section 3.1 in the revised manuscript). The measured El/Ca and associated specimen means and SDs are also being reported in a supplementary table (supplementary table S4). Furthermore, we included statistical analyses of both El/Ca means and the SDs for each species and included environmental parameters (MLD, temperature, salinity and pH) as shown by new 'Spearman correlation matrix' (revised manuscript figure 9 and supplementary figure S12; see 'Figure 1' below). The new spearman correlation matrix of SD shows that environmental parameters, such as MLD, correlate to ICV in some species (i.e. T. clarkei 'big'). 2) in the revised manuscript timeseries plots (figures 3-7) we show the pooled means and SD for each chamber at each given time interval (e.g., see Mg/Ca in Figure 2 below; in the revised manuscript it is titled Figure 3). As reported in our original version in supplementary figure S11, F0 El/Ca was found to be generally lower for G. ruber (figure 3 in the revised manuscript) but generally equal to or higher than the other chambers for T. clarkei, but within 2SD. We also find that during water column mixing (March-May) and deepening of the MLD the SD is higher compared to the other months of the year which is supported by the spearman correlation matrix of SD. Section 4.1, lines 576-589, were included in the revised manuscript: "In most element/Ca ICV is higher during water column mixing months (March-May; e.g., Al/Ca, B/Ca, Ba/Ca, Co/Ca, Fe/Ca, Mg/Ca) in all water depth horizons for T. clarkei 'big' and T. clarkei 'encrusted' and mainly in the two upper water depth horizons (i.e., 120 m and 220 m) for G. ruber albus. These elevated values and high ICV likely reflect the changes in the water properties like the temperature, salinity, pH and nutrient availability derived from the mixing of the water column (Fig. 9, Figs 3-7 panels h, p and x). For some element/Ca ratios (e.g., Na/Ca, Fig. 6/panels g, o and w; Ba/Ca, Fig. 7/panels g, o and w), ICV varies with depth and shows seasonal differences i.e., less variation with depth during water column stratification and more variation with depth during water column mixing; whereas for others (e.g., B/Ca, Fig. 5/panels g, o and w; Sr/Ca, Fig. 4/panels g, o and w) it remains relatively constant with depth.").* As for the original figure 3 in the previous version of the manuscript, this is a summary for comparing all the El/Ca data and

*has been left in the paper but moved to be figure 8 (in the revised manuscript) following restructuring of the results and discussion.*

[Figure]

*Figure 1: Spearman Correlation Matrix for SD of El/Ca (Supplementary figure S12 in revised manuscript).*

[Figure]

*Figure 2: (top panels) MLD, Mg/Ca depth-timeseries (left column: G. ruber, middle: T. clarkei (big), right: T. clarkei (encrusted)), Mg/Ca chamber totals, (bottom panels) and RDA.*

**Item 2:** **This will also require a full report on some basic metrics:**
   **- how many specimens and how many chambers were analyzed?**
   **- What was the variability within ablation profiles? Etc.**

*Reply 2*: In the revised paper we have reported the number of individuals (G. ruber =57, T. clarkei (big)=52, T. clarkei (encrusted)=48) and chambers (measurements; G. ruber = 168, T. clarkei (big)=242, T. clarkei (encrusted)=204) that were analyzed (revised methodology chapter under section 2.2). We include details of the variability within the ablation profiles (section 2.3 in the revised manuscript): The average element-to-calcium ratio from the spot derived LA-ICP-MS count data was calculated from count data immediately after the start of the ablation peak apex until the point identified as the termination of calcite based on the Mg/Ca profile. This time interval represents the stable internal material of the shell; excluding the noisy beginnings and ends of the ablation event. For *G. ruber* the mean ablation time length used for calculation was 4.9±2.3 secs, while for the smaller thinner *T. clarkei* it was 2.6±1.5 secs and 2.4±1.4 secs, for 'big' and 'encrusted' types, respectively."

**Item 3:** **Much of the current Results is spent on differences in time for each of the water depth. But the patterns are very similar, so instead of repeating the results for the different water depths, I suggest to systematically answer the type of/ some of the questions I listed above and illustrate those with new figures.**

*Reply 3:* The time-series figures have been revised to include additional panels showing the averaged values of all depths for each chamber and total mean values with SD (see Figure 2 here for example and our reply to Item 1; now Figures 3-7 in the revised manuscript).

**Item 4:** **Including the MLD in figures 4-8 is confusing, at least in this way. It is the same for every panel. Maybe it works to include it as a color for when a sediment trap is above, and another color for when it is below the MLD. Hope I am making myself clear: the two colors would alternate within a panel and also be different for the different depths (bur obviously remain the same for the three taxa. It may even be sufficient to include that information just for *G. ruber*.**

*Reply 4:* You raise a very good point here and in the revised figures 3-7 we address how the MLD is shown by including separate panels on top of each PF species column for better visuality. However instead of superimposing when and how the sediment traps are within or below the MLD, we carried out statistical analyses of the MLD with relation to the El/Ca changes and SD of the El/Ca, as shown by the Spearman correlation matrices. We also take your suggestion for Redundancy Analysis (RDA) and included a statistical test with the MLD (see also our reply to Item #5 below). It is important to note that in the revised version of the manuscript the MLD has been recalculated to a higher depth resolution, using a different method which shows slightly different trends in comparison to the previous original manuscript (included description of method in the revised manuscript methods section).

**Item 5:** **There is a surprising lack of statistical analysis, while the data allows for comparison along all kinds of dimensions (species, chambers, depths, etc.), which I therefore strongly encourage.**

   a- **The Spearman correlation matrix (figure 9, where the elements should not be near the tick marks between the squares, btw) may not be very useful here: the preceding figures show that the behavior between element in the F-chamber,**

for example, is very similar. I find it interesting that on that level, some of the elements behave very similar (e.g. Mg and Sr), which is lost in the larger comparison of the correlation matrix.
   b- **To disentangle the effect of the different parameters (species, depth, core top or trap, time, MLD, etc.) on the El/Ca and similarity between elements, an RDA may be more appropriate. This would also require rearrangement of section 4.2.**

*Reply 5: Thanks for this comment. As for statistical comparison between chambers please see our response to item #1. We thank you for your very important suggestion to apply RDA which we used to investigate the relationship between pooled mean element/Ca over given depths per species and correlation with environmental variables. We included RDA per species for El/Ca means for each depth and total means (bottom panels in figures 3-7 in the revised manuscript; see figure 2 here for example). In addition to MLD we included additional environmental parameters such as temperature, salinity and pH in these analyses. Given that the RDA assumes linear responses of El/Ca to environmental variables, we choose to include the Spearman correlation matrix as well which has the additional advantage for investigating non-linear correlation between variables and can aid in synthesizing large amounts of data. It appears that the main findings from the RDA and spearman correlation matrices complement each other. For example, in G. ruber albus' Mg/Ca there is good correlation with Temperature, salinity and pH (as also shown in the respective RDA biplot), while for most El/Ca in T. clarkei 'big' the MLD depth is highly correlative (see figure 3 below; Figure 9 in the revised manuscript).*

[Figure]

*Figure 3: Spearman Correlation Matrix for mean El/Ca.*

**Item 6:** **The global compilation (section 5.3) is out of place. Here, all kinds of species are lumped, as well as types of analysis, seasons, etc. It takes a whole other approach to summarize this data and look for meaningful patterns. In the current version of this manuscript, it is also not clear what the overall goal of this comparison is and therefore it is not logically related to the Results and the rest of the Discussion.**

*Reply 6: Thanks for your suggestion. We consider this figure as an important addition to the manuscript, which provides a wide, global-scaled context to the new data reported here from*

*the GOA. We chose to compile this data despite the fact that different El/Ca were measured in regions, and not necessarily on the same species, or the same measuring method; while the data consistency in this compilation can clearly be improved in future, we believe it nevertheless provides important constraints on the interpretation of our new data, as well as previously published data. Therefore, we prefer to keep figure 12.*

**Item 7: (in the pdf file)**
**Line 332: "This may be a good reason to illustrate this with some LA profiles and what the presence of such crust does to the length of the profiles/ heterogeneity within profiles? Is the crust equally thick across depths?"**

*Reply 7: Thank you for this point. Regarding the length of the ablation: the mean ablation time for T. clarkei 'encrusted is 2.6 sec (with SD=1.5 sec) and for T. clarkei 'big' the mean ablation time is 2.4 sec (with SD=1.4 sec), meaning the difference is not great and falls within the error. While we agree that examining the heterogeneity of the ablation profiles of the crust will add much valuable information it is beyond the scope of this manuscript. As for whether the crust is equally thick across depths: we do not have measurements of whole shell thickness or just crust thickness in all the individuals we measured. During the laser ablation measurement, we observed that in T. clarkei 'encrusted' the dwelling time from the start of the ablation until a hole appeared may change between specimens from different depths; for example, in the individuals from September: the ablation dwelling time was between 2-4 seconds in 220 m and between 4-12 seconds in 570 m. Nevertheless, this observation has not been statistically tested nor does it necessarily reflect the thickness of the crust itself as it measures both layers (crust and internal layers) of the test. However, this question will be very interesting to investigate in a follow-up study.*

**Item 8: (in the pdf file)**
**Line 412: "it would be nice if the authors can say something about the possible causes for the deviation of El/Ca:**
- **Shorter ablation time affect the El/Ca average?**
- **Presence of coating skewing the relatively thin wall of F0?**
- **Biomineralization processes?**
- **Maybe something else?**

*Reply 8: For both G. ruber albus and T. clarkei the final chamber is systematically different from the previous ones. But, while in G. ruber albus F0 is usually lower than the previous chambers in the same specimen, in T. clarkei it is usually higher.*
- *We do not think that the lower calculated temperatures from F0 in G. ruber are due to shorter ablation time as T. clarkei has a much shorter ablation time than G. ruber but still has higher F0 values.*
- *We also do not believe it's the presence of a coating as G. ruber is not known for having a secondary crust at all. Furthermore, T. clarkei 'encrusted' which is coated with crust also has high F0 values compared to G. ruber which would have resulted in higher calculated temperatures (if it was suitable for reconstructing temperatures). If you are referring to coating of other materials (like glue for example or something which is found in the water), we don't think it would cause the deviations of El/Ca as it should have had the same effect on both G. ruber albus and T. clarkei and not show two different systematics.*

- *The chamber differences could be related to the biomineralization processes, which would probably be species specific. Unfortunately, to check this is beyond the scope of this manuscript.*
- *Another possibility is the migration of G. ruber albus deeper in the water column while it calcifies its final chamber. However, we are not able to examine this option with the resolution of sediment traps that we have (every ∼100 m).*

**Item 9:** **(in the pdf file)**
**Line 443-444: "but the variability in time in B/Ca does not match the variability in $p$H over time. This argues against such a control!".**

*Reply 9: We see that the trends of the mean values of B/Ca in T. clarkei 'big' and 'encrusted' may match pH at certain depth intervals (350 m – 570 m; Figures 11h, 11i, 11j, 11n, 11o in revised manuscript). We clarified this in the revised manuscript (lines 743-751).*

**Item 10:** **(in the pdf)**
**Line 335-336: "But the variability among the different chambers for the two morphotypes are not always the same, right? For example, Mg/Ca at 450 meters. If the patterns in El/Ca would be similar between the two T. clarkei morphotypes, they would not have to be shown separately."**

*Reply 10: Indeed, the variability between the two phenotypes are not always the same, as also evident in the statistical differences, which is why they are shown separately.*

**Item 11:** **minor suggestions and corrections in the pdf file**

*Reply 11: All the minor suggestions and corrections raised will be addressed accordingly in the revised manuscript:*

Line 57: *Deleted.*
Line 58: *Accepted.*
Line 59-60: *Accepted, clarified.*
Line 62: *Deleted.*
Line 63: *Noted and clarified. Lines 63-64 in the revised manuscript.*
Line 70: *Yes. We clarified this. Line 75 in the revised manuscript.*
Line 71: *Yes. We clarified this. Line 76-77 in the revised manuscript.*
Line 74: *Deleted.*
Line 75: *Deleted.*
Line 76: *Deleted.*
Line 79: *Deleted.*
Line 113: *Accepted, changed.*
Line 119: *Accepted, changed.*
Line 131-133: *Accepted and clarified (lines 145-150 in the revised manuscript)*
Line 144: *Accepted, clarified.*
Lines 166: *Accepted, changes.*
Lines 167-168: *Accepted, corrected.*
Line 208-209: *Accepted, clarified. Line 531 in the revised manuscript*
Line 216: *Accepted and moved to line 534 in the revised manuscript*
Line 323-324: *Accepted and clarified. Line 595 in the revised manuscript*

Line 325: *Yes, changed accordingly (line 596-598 in the revised manuscript). See also our reply to Item #1.*

Sincerely,

Noy Levy on behalf of all co-authors